# Substantiating freedom from parasitic infection by combining transmission model predictions with disease surveys

Edwin Michael[1], Morgan E. Smith[1], Moses N. Katabarwa[2], Edson Byamukama[3], Emily Griswold[2], Peace Habomugisha[3], Thomson Lakwo[4], Edridah Tukahebwa[4], Emmanuel S. Miri[5], Abel Eigege[5], Evelyn Ngige[6], Thomas R. Unnasch[7] & Frank O. Richards[2]

Stopping interventions is a critical decision for parasite elimination programmes. Quantifying the probability that elimination has occurred due to interventions can be facilitated by combining infection status information from parasitological surveys with extinction thresholds predicted by parasite transmission models. Here we demonstrate how the integrated use of these two pieces of information derived from infection monitoring data can be used to develop an analytic framework for guiding the making of defensible decisions to stop interventions. We present a computational tool to perform these probability calculations and demonstrate its practical utility for supporting intervention cessation decisions by applying the framework to infection data from programmes aiming to eliminate onchocerciasis and lymphatic filariasis in Uganda and Nigeria, respectively. We highlight a possible method for validating the results in the field, and discuss further refinements and extensions required to deploy this predictive tool for guiding decision making by programme managers.

[1] Department of Biological Sciences, University of Notre Dame, Notre Dame, IN 46556, USA. [2] Emory University and The Carter Center, One Copenhill, 453 Freedom Parkway, Atlanta, GA 30307, USA. [3] The Carter Center, Uganda, P.O. Box 12027 Kampala, Uganda. [4] Vector Control Division, Ministry of Health, 15 Bombo Road P.O. Box 1661 Kampala, Uganda. [5] The Carter Center, Nigeria, 1 Jeka Kadima Street off Tudun Wada Ring Road, Jos, Nigeria. [6] Federal Ministry of Health, Federal Sceretariat, Garki-Abuja, Nigeria. [7] Global Health Infectious Disease Research, College of Public Health, University of South Florida, 33620 Tampa, FL, USA. Correspondence and requests for materials should be addressed to E.M. (email: emichael@nd.edu)

A key development in human health over the past decade has been the initiation and implementation of several large-scale disease-specific global health initiatives aiming to achieve the regional or global elimination or eradication of a range of debilitating infectious diseases, including polio, malaria, tuberculosis, and the many parasitic diseases that together constitute the so-called "Neglected Tropical Diseases (NTDs)"[1–4]. These programmes have led to remarkable reductions in the global burdens of these diseases to low or moderately low levels in the tropics and subtropics, but achieving regional or national disease elimination has proved to be far more challenging with increasing appreciation that meeting this exacting goal would require the application of intensified interventions over large areas and over extended periods of time[5]. For such programmes, a major managerial question is deciding when the disease has been eliminated over a spatial domain of interest so that control activities can be stopped. This clearly presents a critical decision point in the management of these programmes, as stopping interventions too soon would lead to the inevitable re-emergence of transmission and reversal of the health gains achieved[6,7], whereas stopping too late would lead to continuing with interventions and monitoring for many more years than necessary.

There are two major problems with developing analytical frameworks for assisting with declaring that a population (whether a village, district, province, or the whole country) is free from a disease. First, as assessments of disease status are normally made by testing samples from a population, the detection of infected hosts is dependent on the characteristics of disease surveys, including the representativeness and size of samples used in surveys, as well as the diagnostic performance (sensitivity and specificity) of the tests used to classify infection status[8]. As very few diagnostic tests are perfect, their use in disease surveys means that it is often possible to obtain some positive test results even if there is no disease present because of the test producing false positives, whereas if the disease is present it is also possible to get false-negative test results and thus miss diseased individuals. These issues mean that it is impossible to prove that a population is free from disease in an area even with large sample sizes, as there is always the chance that an infected individual may have been missed or that the test result is wrong[8,9]. However, although it can never be proved entirely that a population is free from a disease, if enough individuals are surveyed over an area and the performance of the diagnostic test used is taken into account, then it is possible to show how unlikely it is that a population has continuing infection or disease[8–12].

The second problem relates to defining the threshold prevalence of disease that denotes whether transmission is ongoing or has been interrupted in a population[13–17]. This threshold represents the maximum acceptable disease prevalence or the design prevalence that the survey can be expected to reliably identify given its sampling and design characteristics (see Methods)[8,18]. There are two issues connected with setting this level of acceptable infection. First, its specification is important mainly because calculations of the probability of having achieved disease freedom based on a survey are founded on the (hypothetical) null hypothesis that infection is present at the specified design prevalence in the population[19]. Given this, the design prevalence is essentially an abstract statement of the level of infection expected to be present in nature, the reliable measurement of which is used to assess whether transmission is ongoing in a setting. Second, assigning a value to this prevalence implies that if infection is found to be present in a population but below the specified positive design prevalence, then we assume that parasite transmission has been interrupted. This means that its level should

essentially be set based on knowledge of parasite extinction dynamics, particularly with regard to the values of the infection breakpoint thresholds that govern parasite transmission occurrence or absence in a population[6,15–17,20]. Arbitrarily set design prevalences, as traditionally used in the case of livestock disease management or indeed used by the World Health Organization (WHO) for various NTDs (e.g., the 1% microfilariae (Mf) prevalence target for determining lymphatic filariasis (LF) or onchocerciasis elimination[21,22]), will be insufficient in this regard, as such levels may not mean parasite transmission has been interrupted or that there is a high likelihood of attaining zero prevalence once crossed[6]. In other words, assessing freedom from infection requires setting design prevalences that signify the eventual attainment of zero prevalence, i.e., not just low but sustained prevalence. It is noteworthy, in passing, that as a departure from previous freedom from disease calculations[8,18,19,23], the existence of these positive infection breakpoint thresholds implies that parasite freedom calculations can be practically performed even before negative surveillance reports are obtained.

A key difficulty, however, is arriving at the value of this threshold prevalence; recent work in modelling vector-borne macroparasitic transmission dynamics has shown that these breakpoint prevalences are highly dynamical and will invariably vary from site to site depending on local transmission conditions[6,24–27]. This result indicates that tools based on structured infection surveys to make claims of parasitic infection freedom must crucially integrate knowledge regarding the population ecology of breakpoints in different localities together with survey metrics (sample size, diagnostic test performance) to support substantiations of infection elimination in host populations.

In this study, we present and demonstrate a new quantitative approach that facilitates the coupling of surveillance data from surveys designed to monitor or track the impact of interventions on infection trends in communities with parasite transmission model (PTM)-based estimates of infection breakpoint values for predicting the probability of achieving infection freedom in a setting. The approach is founded on the principles underlying the use of structured surveys to demonstrate the achievement of freedom from disease or infection in treated populations, and the use of data-driven parasite transmission modelling as a tool for predicting the infection breakpoint prevalences that will need to be used along with infection surveys for quantifying the likelihood or probability of meeting the goal of parasite transmission interruption. We present the construction of a novel integrated parasite freedom from infection (PFFI) tool to facilitate this coupling of two predictive tools, viz. data-driven PTMs and survey-based proof-of-freedom methods, for enabling the making of these calculations. We illustrate the utility of this modelling approach to support declarations of parasitic infection freedom by applying the tool to actual surveillance data from the current programmes to eliminate onchocerciasis and LF in Uganda and Nigeria, respectively. In particular, we show how the method can be applied to quantify infection freedom and validate its successful attainment based on data from active ongoing longitudinal surveys tracking changes in infection prevalence in sentinel sites. The importance of the integrated PFFI methodology for supporting sample size calculations will also be noted in this exercise. We end by discussing the critical need for evaluating such frameworks and present a means to do so via the use of sequential vector sampling methodologies. We also describe how our framework may be extended further to estimate cumulative evidence of freedom from serial data and to facilitate the making of area-wide PFFI estimations.

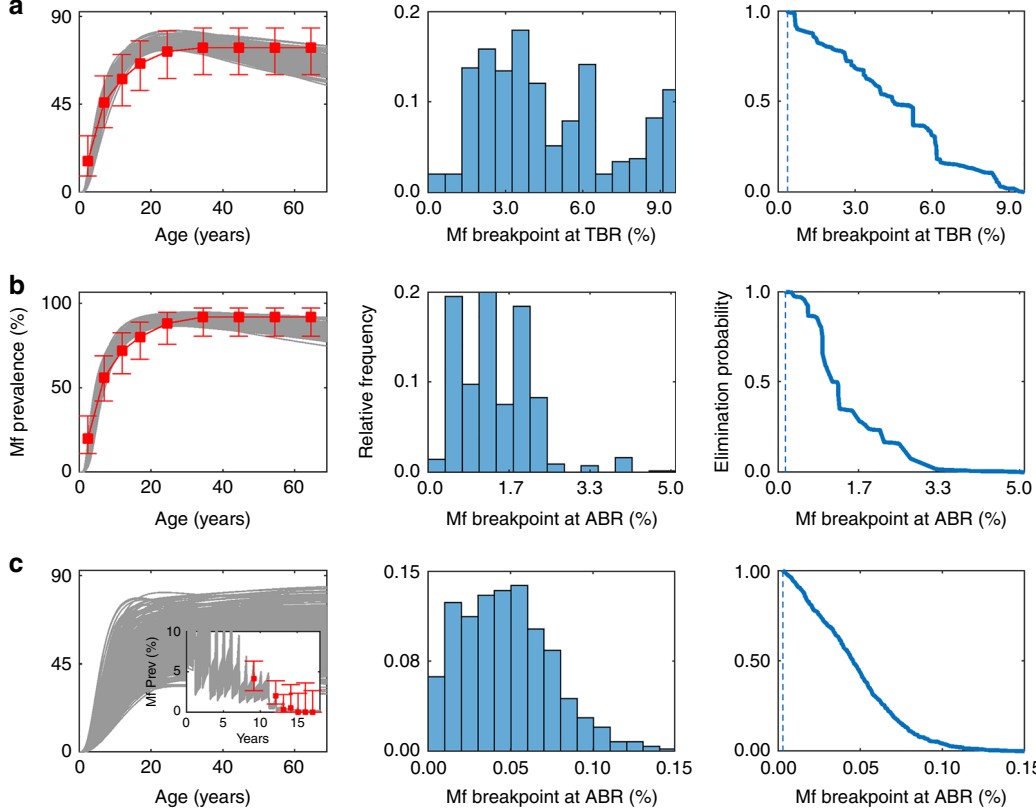

**Fig. 1** Model fits and estimated transmission breakpoints. The model fits (grey curves) to baseline microfilariae prevalence from two onchocerciasis endemic sites, **a** Buriri, Uganda and **b** Masaloa, Uganda, and one LF endemic site **c** Gbuwhen, Nigeria, are shown. For Buriri and Masaloa, age-stratified Mf prevalence patterns (shown in the figure as red squares for estimated plateau-type patterns with error bars representing the 95% binomial confidence intervals) used for fitting were constructed according to the reported community-level Mf prevalence (Tables 1, 2). For Gbuwhen, the model was fit to post-intervention data (shown in the inset plot as red squares with error bars representing the 95% binomial confidence intervals) and the baseline curves were hindcasted. The distribution of the model-calculated Mf breakpoints (centre) and the corresponding inverse ECDFs (right) are shown for sites **a–c**. The vertical dashed lines in the inverse ECDF plots denote the Mf breakpoint values corresponding to the 95% elimination probability thresholds applicable in each village

## Results

**Model fits and breakpoints.** Figure 1 provides an illustration of the ability of our data-driven modelling approach to both identify best-fitting onchocerciasis and LF models for describing observed disease-specific Mf prevalence data in different community settings, and to facilitate the estimation of the corresponding Mf transmission thresholds for use as design prevalences (see Methods). Results are shown for two of the onchocerciasis and one of the LF study sites in the figure, with the full set of model fits and the applicable 95% Mf elimination probability thresholds (EPTs)—the Mf prevalence values, which when crossed below would signify a 95% probability that transmission interruption is likely to occur[6,25]—in all the other study sites given in Supplementary Fig. 1 and Supplementary Fig. 2. As described in detail in Singh and Michael[6], and Michael and Singh[25], these EPTs were derived using an inverse empirical cumulative density function (ECDF) approach and are calculated at either the annual biting rate (ABR) or the threshold biting rate (TBR), depending on whether a particular location has implemented vector control measures in addition to mass drug administration (MDA). Empirical evidence for the existence of these model-derived thresholds, including validation that crossing below such thresholds would lead to non-detectable filarial transmission, is detailed in Reimer et al.[28] and indicate that the breakpoints predicted in this study are not merely theoretical outputs arising

from analyses of our data-fitted models but are very likely to reflect actually occurring natural phenomena in the field.

The numerical values of the Mf 95% EPTs for each of the onchocerciasis and LF study sites are further listed in Tables 1, 2 respectively. It is noteworthy that for the onchocerciasis sites, these breakpoint estimates were derived via analyses of the onchocerciasis models that were found to best fit the joint baseline Mf prevalence and ABR data measured directly in each site. Mf breakpoint values are shown at the TBRs for Mt. Elgon villages and at the prevailing ABR intensity for all sites belonging to the Madi Mid North focus in Uganda. As pointed out by Singh and Michael[6], Mf breakpoints at TBR are applicable when vector control is added to MDA and so constitute the relevant target breakpoints in the case of Mt. Elgon, whereas in the case of MDA-only interventions as carried out in the Madi Mid North focus, the corresponding breakpoints of interest are those that are applicable at the prevailing and undisturbed ABR rates. In the case of some of the LF study sites, however, the site-specific breakpoint values were estimated by carrying out analyses based on age-prevalence and ABR values hindcasted to pre-control baseline states using the models that best fitted the available post-intervention prevalence data in each site[26]. This was done to account for the fact, first, that although all the LF sites received MDA with Ivermectin and Albendazole starting from 2000 and the first Mf survey was done just before this intervention[29],

**Table 1 Onchocerciasis survey data for Ugandan villages and model-predicted Mf prevalence thresholds at village-specific ABR and TBR**

| Focus (transmission status) | Village | Year | Total population | Sampled | No. positive | Mf prevalence (%) | Mf breakpoint (design prevalence) | |
|---|---|---|---|---|---|---|---|---|
| | | | | | | | 95% EP threshold at ABR | 95% EP threshold at TBR |
| Mt. Elgon (interrupted) | Bubungi | 1994[a] | 601 | – | – | 75 | – | 0.47 |
| | | 2005 | 442 | 156 | 3 | 1.92 | | |
| | | 2011 | 528 | 107 | 0 | 0 | | |
| | Bunabutiti | 1994[a] | 1098 | – | – | 53.8 | – | 0.36 |
| | | 2005 | 177 | 110 | 0 | 0 | | |
| | | 2011 | 650 | 123 | 1 | 0.81 | | |
| | Bunambatsu | 1994[a] | 969 | – | – | 58.8 | – | 0.42 |
| | | 2005 | 1127 | 124 | 2 | 1.61 | | |
| | | 2011 | 951 | 133 | 1 | 0.75 | | |
| | Buriri | 1994[a] | 350 | – | – | 61.3 | – | 0.68 |
| | | 2005 | 426 | 137 | 1 | 0.73 | | |
| | | 2011 | 544 | 63 | 1 | 1.59 | | |
| Madi Mid North (ongoing) | Andra | 1993[a] | 510 | – | – | 70 | 0.15 | – |
| | | 2004 | 698 | 101 | 7 | 6.9 | | |
| | Madulu | 1993[a] | 1620 | – | – | 72 | 0.15 | – |
| | | 2004 | 812 | 99 | 3 | 3 | | |
| | | 2011 | 1182 | 106 | 2 | 1.9 | | |
| | Masaloa | 1993[a] | 1122 | | | 76 | 0.11 | – |
| | | 2004 | 1109 | 129 | 5 | 3.9 | | |
| | Palaure | 1993[a] | 214 | – | – | 100 | 0.03 | – |
| | Pacunaci | 2004 | 430 | 88 | 12 | 13.6 | | |

In Mt. Elgon, *Simulium neavei* is the vector species responsible for transmission[67], whereas *Simulium damnosum* is responsible in Madi Mid North[68]
[a]Pre-intervention survey

several of these sites had also previously received annual MDA with Ivermectin only under the Nigerian Onchocerciasis programme from the year 1993 (Table 2). Second, information on individual baseline Mf prevalence or ABR values before the onchocerciasis intervention were missing for these sites. The actual values of Mf breakpoints displayed in Tables 1, 2 corroborate our previous modelling findings[6,24,25] regarding filarial breakpoints, viz. they will invariably: (1) vary significantly between endemic communities, (2) be higher at TBR, and (3) demonstrably be much lower than the WHO suggested threshold of 1% Mf prevalence for use as intervention targets for either infection. The results also suggest the added intriguing possibility that these Mf thresholds applicable to LF could also be significantly lower (at least by onefold) than those that may apply in communities for onchocerciasis (Tables 1, 2).

**Applying the PFFI tool**. Tables 3, 4 presents the results of applying the PFFI tool (Fig. 2) to the longitudinal Mf prevalence data as reported in both the onchocerciasis and LF sites investigated in this study (Tables 1, 2). The results given in Table 3 for the onchocerciasis dataset indicate that in the case of the Mt. Elgon sites, except for the village of Buriri in 2011, all villages were found to have very low probabilities for accepting the null hypothesis, indicating a corresponding high probability that each population is free from infection at confidence levels generally above 95%. A noteworthy feature of the results, however, was that high confidences of having achieved infection freedom (> 95% confidence) were already apparent by 2005 in each of the Mt. Elgon sites (Table 3), and that, in general, the confidence of having achieved infection freedom improved further by 2011, as would be expected given that interventions continued in these sites after parasite elimination thresholds were first crossed in 2005. This result also suggests that since the decision to stop interventions in Mt. Elgon

was made only in 2011 based on the criteria noted above, a further 5 years of unnecessary interventions may have been implemented in these sites. In the case of the Madi Mid North focus, however, the results indicate that except for Palaure Pacunaci village, none of the other villages may be deemed to have eliminated the infection (Table 3). Although this conclusion appears to support the decision taken by the Uganda Onchocerciasis Elimination Programme to continue MDA in this focus, our analysis indicates that the results reported in Table 3 for these villages could be a direct outcome of the inadequate sample sizes used for measuring infection prevalences in the respective populations (Table 1).

The results from the corresponding PFFI analysis for the Nigerian LF sites are shown in Table 4. As in the case for the onchocerciasis sites, these results also indicate that if sufficient sample sizes were used for assessing infection levels (Table 2), freedom from infection could have been declared in the present LF sites and interventions stopped much earlier than was done in practice.

**Impact of sample sizes and diagnostic test characteristics**. The impact of sample sizes for calculating confidences of having achieved infection freedom is highlighted by the results shown in Table 5 for the inconclusive onchocerciasis sites investigated in this study. The results show that if sample sizes were increased from the original numbers while maintaining the actual Mf prevalences measured and the design prevalences estimated in each of these sites, the increased sample sizes would have enabled the rejection of the null hypothesis and therefore support the conclusion that each of these populations was free of infection as early as the year 2004 in the case of Madulu and Masaloa villages in the Madi Mid North focus, and by year 2011 in the case of Buriri village in Mt. Elgon (Table 5). This finding indicates the importance of deriving and using adequate sample sizes when

**Table 2 LF survey data for Nigerian villages and corresponding model-predicted Mf prevalence thresholds at ABR**

| Nigerian state | Village | Year | Total population[a] | No. sampled | No. positive | Mf prevalence (%) | Mf breakpoint (design prevalence, 95% EP threshold at ABR) |
|---|---|---|---|---|---|---|---|
| Nasarwa | Gbuwhen[b] | 2001 | 1503 | 508 | 19 | 3.7 | 0.014 |
| | | 2004 | | 446 | 8 | 1.8 | |
| | | 2005 | | 286 | 1 | 0.3 | |
| | | 2006 | | 183 | 1 | 0.5 | |
| | | 2007 | | 196 | 0 | 0.0 | |
| | | 2008 | | 127 | 0 | 0.0 | |
| | | 2009 | | 175 | 0 | 0.0 | |
| | Maiganga[b] | 2001 | 431 | 486[c] | 23 | 4.7 | 0.012 |
| | | 2005 | | 169 | 5 | 3.0 | |
| | | 2006 | | 126 | 7 | 5.6 | |
| | | 2007 | | 158 | 1 | 0.6 | |
| | | 2008 | | 109 | 2 | 1.8 | |
| | | 2009 | | 152 | 1 | 0.7 | |
| Plateau | Dokan Tofa | 2002[d] | 2677 | 419 | 21 | 5.0 | 0.016 |
| | | 2007 | | 151 | 2 | 1.3 | |
| | | 2008 | | 158 | 0 | 0.0 | |
| | | 2009 | | 223 | 1 | 0.4 | |
| | Gwamlar | 2002[d] | 6052 | 494 | 33 | 6.7 | 0.017 |
| | | 2006 | | 240 | 29 | 12.1 | |
| | | 2007 | | 128 | 2 | 1.6 | |
| | | 2008 | | 100 | 5 | 5.0 | |
| | | 2009 | | 143 | 7 | 4.9 | |
| | Lankan[b] | 2001 | 2119 | 274 | 9 | 3.3 | 0.012 |
| | | 2004 | | 365 | 27 | 7.4 | |
| | | 2005 | | 243 | 11 | 4.5 | |
| | | 2006 | | 81 | 2 | 2.5 | |
| | | 2007 | | 117 | 2 | 1.7 | |
| | | 2008 | | 173 | 7 | 4.0 | |
| | | 2009 | | 201 | 0 | 0.0 | |
| | Piapung | 2002[d] | 2068 | 403 | 40 | 9.9 | 0.016 |
| | | 2007 | | 187 | 18 | 9.6 | |
| | | 2009 | | 291 | 6 | 2.1 | |
| | Seri[b] | 2003 | 1207 | 527 | 56 | 10.6 | 0.012 |
| | | 2005 | | 321 | 5 | 1.6 | |
| | | 2006 | | 157 | 2 | 1.3 | |
| | | 2007 | | 133 | 1 | 0.8 | |
| | | 2008 | | 110 | 3 | 2.7 | |
| | | 2009 | | 258 | 0 | 0.0 | |

[a]Total population estimate calculated by dividing the 2009 eligible population given in Richards et al.[30] by the fraction of the population eligible for treatment. We assume the eligible population is equal to the population of age greater than 5 years, which was calculated to be 0.8125 in Nigeria
[b]Denotes a village where no pre-intervention data were available so baseline conditions were hindcasted by fitting the model to post-intervention survey data
[c]As the sample size was greater than the estimated population size, the entire population was considered to have been sampled
[d]Pre-intervention survey
Although both Culex and Anopheles mosquitoes are present in this region, Anopheles is the primary vector responsible for transmission[30]

**Table 3 Results from the onchocerciasis PFFI analyses**

| Focus (transmission status) | Village | Year | Probability of null hypothesis, $P_O$ | Probability of alternative hypothesis, $P_a$ | Confidence of freedom | Classification[a] |
|---|---|---|---|---|---|---|
| Mt. Elgon (interrupted) | Bubungi | 2005 | 0.021 | 0.986 | 0.979 | Y |
| | | 2011 | 0.002 | 1.000 | 0.998 | Y |
| | Bunabutiti | 2005 | 0.001 | 1.000 | 0.999 | Y |
| | | 2011 | 0.008 | 0.998 | 0.992 | Y |
| | Bunambatsu | 2005 | 0.033 | 0.987 | 0.967 | Y |
| | | 2011 | 0.005 | 0.999 | 0.995 | Y |
| | Buriri | 2005 | 0.003 | 0.999 | 0.997 | Y |
| | | 2011 | 0.120 | 0.961 | – | Insufficient evidence |
| Madi Mid North (ongoing) | Andra | 2004 | 0.807 | 0.241 | – | Insufficient evidence |
| | Madulu | 2004 | 0.157 | 0.878 | – | Insufficient evidence |
| | | 2011 | 0.048 | 0.971 | 0.952 | Y |
| | Masaloa | 2004 | 0.310 | 0.778 | – | Insufficient evidence |
| | Palaure Pacunaci | 2004 | 0.999 | 0.001 | 0.001 | N |

[a]N not free from infection, Y free from infection

**Table 4 Results from the LF PFFI analyses**

| Nigerian state | Village | Year | Probability of null hypothesis, $P_O$ | Probability of alternative hypothesis, $P_a$ | Confidence of freedom | Classification[a] |
|---|---|---|---|---|---|---|
| Nasarwa | Gbuwhen | 2001 | 0.100 | 0.925 | – | Insufficient evidence |
| | | 2004 | 0.000 | 1.000 | 1.000 | Y |
| | | 2005 | 0.000 | 1.000 | 1.000 | Y |
| | | 2006 | 0.001 | 1.000 | 0.999 | Y |
| | | 2007 | 0.000 | 1.000 | 1.000 | Y |
| | | 2008 | 0.001 | 1.000 | 0.999 | Y |
| | | 2009 | 0.000 | 1.000 | 1.000 | Y |
| | Maiganga | 2001 | 0.429 | 0.579 | – | Insufficient evidence |
| | | 2005 | 0.120 | 0.928 | – | Insufficient evidence |
| | | 2006 | 0.666 | 0.443 | – | Insufficient evidence |
| | | 2007 | 0.002 | 1.000 | 0.998 | Y |
| | | 2008 | 0.072 | 0.975 | – | Insufficient evidence |
| | | 2009 | 0.003 | 1.000 | 0.997 | Y |
| Plateau | Dokan Tofa | 2007 | 0.017 | 0.996 | 0.983 | Y |
| | | 2008 | 0.000 | 1.000 | 1.000 | Y |
| | | 2009 | 0.000 | 1.000 | 1.000 | Y |
| | Gwamlar | 2006 | 1.000 | 0.000 | 0.000 | N |
| | | 2007 | 0.042 | 0.989 | 0.958 | Y |
| | | 2008 | 0.613 | 0.564 | – | Insufficient evidence |
| | | 2009 | 0.573 | 0.577 | – | Insufficient evidence |
| | Lankan | 2001 | 0.112 | 0.933 | – | Insufficient evidence |
| | | 2004 | 0.981 | 0.029 | 0.019 | N |
| | | 2005 | 0.429 | 0.674 | – | Insufficient evidence |
| | | 2006 | 0.218 | 0.917 | – | Insufficient evidence |
| | | 2007 | 0.062 | 0.982 | – | Insufficient evidence |
| | | 2008 | 0.352 | 0.766 | – | Insufficient evidence |
| | | 2009 | 0.000 | 1.000 | 1.000 | Y |
| | Piapung | 2007 | 0.997 | 0.006 | 0.003 | N |
| | | 2009 | 0.008 | 0.997 | 0.992 | Y |
| | Seri | 2003 | 1.000 | 0.000 | 0.000 | N |
| | | 2005 | 0.001 | 1.000 | 0.999 | Y |
| | | 2006 | 0.012 | 0.997 | 0.988 | Y |
| | | 2007 | 0.008 | 0.999 | 0.992 | Y |
| | | 2008 | 0.185 | 0.917 | – | Insufficient evidence |
| | | 2009 | 0.000 | 1.000 | 1.000 | Y |

[a]N not free from infection, Y free from infection

performing parasite infection freedom assessments, because inadequate sample sizes would indicate ongoing transmission, as was deemed for Madi Mid North, when in reality a high probability may have been reached indicating that transmission has ceased.

An important caveat to note is that these results are highly sensitive to the specificity of the diagnostic tests used. If the specificity of a test tends to 1 then the power of declaring a population infection-free will decline dramatically for a given sample size (Fig. 3), essentially because of significant reductions in the observation of false-positive diagnostic test results. Only by increasing sample sizes at high test specificities for a given combination of test sensitivity and design prevalence values will high survey confidences (e,g., > 95%) be achieved for declaring infection freedom in such a case (Fig. 3). This result underscores the paramount importance of using appropriate test performance values and sample size calculations when carrying out PFFI predictions. Note, because there is a lack of clear consensus on the sensitivity and specificity of the diagnostic tools used in these surveys (thick blood smear for LF and skin snip microscopy for onchocerciasis)[30–36], we used a neutral set of test performance values for both these Mf detection tests, viz. Se = 0.95 and Sp = 0.95, in the PFFI calculations described here. This means that the results in Tables 3, 4 need to be seen as indicative rather than definitive at this stage, although, as we show in Supplementary Table 1 (and corroborating the results depicted in Fig. 3),

lowering Se to 0.85 while keeping Sp at 0.95 will make essentially no difference to the present results.

## Discussion

This work shows that demonstrating freedom from parasitic infection due to applications of community-wide interventions in a population essentially requires calculating the probability that a particular surveillance plan will reliably detect a transmission interrupting threshold in that population. In particular, we have shown that when parasite breakpoint values are used as design prevalences and a surveillance system finds infection below these prevalences at an acceptably high level of confidence (e.g., 95%), then we can be sufficiently confident that transmission interruption has occurred. This incorporation of positive-valued breakpoint prevalence values in the freedom calculations by the present predictive framework sets it apart from the traditional application of the freedom from disease approach, in which design prevalences are arbitrarily set and negative surveillance data are used to carry out infection freedom calculations (see a recent review of these approaches by Stresman et al.[37]). Although many questions still remain to be resolved in the actual application of these methods to human diseases (see below and ref. [37]), the work presented here, together with results from an analysis carried out by Dukpa et al.[38] with respect to validating the reported status of a district in Bhutan being free from foot-and-

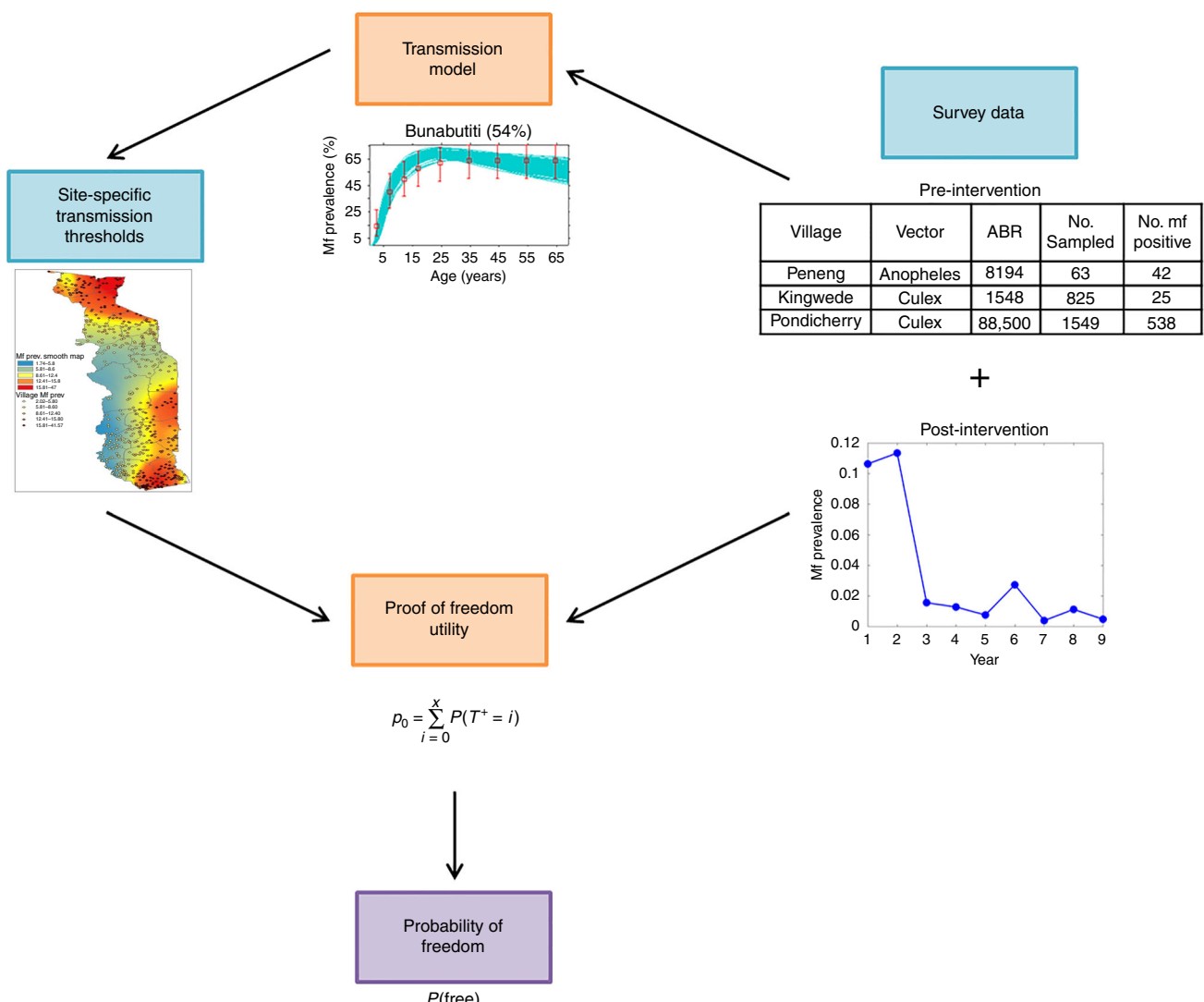

**Fig. 2** Integrated parasite freedom from infection tool. The relationship between the two predictive computational tools (orange boxes), site-specific inputs (blue boxes), and the output quantifying the probability of freedom (purple box) is displayed graphically. The pre-intervention survey data inform the calibration of the transmission model, whereas the estimated site-specific Mf prevalence thresholds and post-intervention survey data informs the freedom from infection calculations. The images included in this figure are illustrative in nature and the data shown are not pertinent to this study. A complete set of pre-intervention survey data used for calibration the model includes the site location, the vector species responsible for transmission, the prevailing annual biting rate, and the baseline mf survey results (data from [6]). The map of site-specific transmission thresholds highlights the spatial heterogeneity of breakpoints used here for defining the design prevalence in the PFFI calculations (data from [69,70])

### Table 5 Required sample sizes and corresponding PFFI classifications

| Focus (transmission status) | Site | Year | Survey sample size | Required sample size | Probability of null hypothesis, $P_0$ | Probability of alternative hypothesis, $P_a$ | Confidence of freedom | Classification[a] |
|---|---|---|---|---|---|---|---|---|
| Mt. Elgon (status: interrupted) | Buriri | 2011 | 63 | 93 | 0.031 | 0.992 | 0.969 | Y |
| Madi Mid North (status: ongoing) | Andra | 2004 | 101 | 401 | 0.961 | 0.049 | 0.039 | N |
| | Madulu | 2004 | 99 | 359 | 0.044 | 0.972 | 0.956 | Y |
| | Masaloa | 2004 | 129 | 889 | 0.048 | 0.959 | 0.952 | Y |

[a]N not free from infection, Y free from infection

mouth disease, point to the potential power of applying these non-zero prevalence thresholds for guiding the making of defensible evidence-based intervention stopping decisions in disease control programmes.

The PFFI approach developed and used in this study also differs conceptually and methodologically from existing WHO infection freedom assessment frameworks. First, it explicitly addresses the problem related to the use of arbitrarily defined

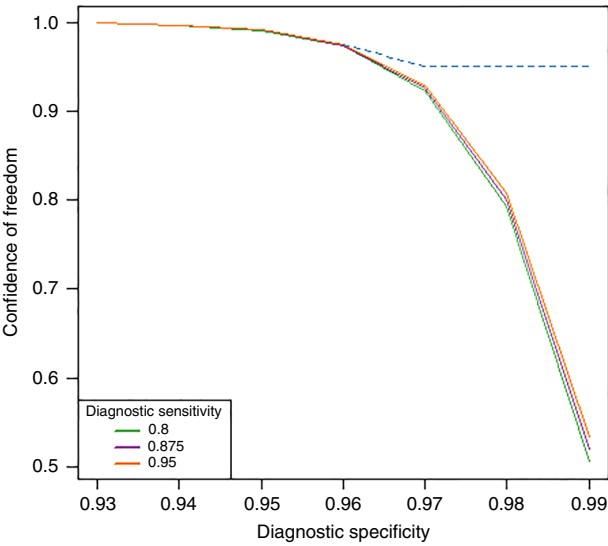

**Fig. 3** Power of a survey to declare infection freedom in a population. The solid lines depict the change in the power of an infection freedom survey (in the form of confidence for declaring freedom) as the specificity of the diagnostic test used increases to 0.99 for three fixed sensitivity values (Se = 0.8, 0.875, and 0.95) and fixed survey sample sizes. The results show the dramatic decline in the power of a survey to declare infection freedom as diagnostic test specificity increases from 0.95 towards perfect specificity, whereas increases in sensitivity for a given specificity value has only a slight effect. The dotted line shows that the power of the survey can be maintained even at high specificities by changing the sample size. As test specificity increases (for a given sensitivity), confidence of a survey to declare infection freedom can only be maintained at high levels (> 95%) by increasing sample sizes significantly. In the present example, which is based on the 2011 survey results from Bunabutiti (Table 1), the required sample sizes to maintain this high confidence for declaring freedom from onchocerciasis (dotted line) work out to be 123 at Sp = 0.96 to as high as 650 at Sp = 0.99

infection thresholds for assessing the achievement of parasite transmission interruption by existing methods[21,39–41]. We have shown using modelling studies, how, in this regard, if arbitrarily set infection thresholds are used as elimination targets for a disease, such as LF, then if the actual breakpoint thresholds are not crossed, such an approach will invariably result in a high likelihood that resurgence of infection will occur[6]. Empirical results have begun to support these theoretical findings; for example, a recently reported community trial of MDA versus combined MDA and vector control interventions against LF in India has provided the first field evidence for this potential for resurgence of infection if MDA is stopped before these natural breakpoints are crossed[7]. A similar explanation may also underlie the findings from a recently concluded post-intervention assessment carried out across Sri Lanka, which showed the persistence of LF transmission in many communities that had both undergone MDA interventions and met the standard WHO LF endpoint criteria[40]. These results reinforce a key conclusion from both our original modelling and present works, viz. if reliable declarations of transmission interruption are to be made in parasite elimination programmes, there is a vital need to apply ecologically sound breakpoint values rather than use untested, arbitrarily defined target thresholds for making such decisions[6,25].

Second, current WHO disease freedom assessment tools are largely developed and applied with little consideration paid to the impacts that sample sizes and diagnostic tool statistics can have in the evaluation of infection endpoints in a population[21,39–41]. Our

analysis has highlighted the vital role that the performance of diagnostic tools can play not only for reliably clarifying infection levels that can be expected to occur in a population, but also for calculating the sample sizes required for carrying out dependable freedom from infection assessments. This is a well-established consequence of using structured surveys for detecting infection[8,9,11,18,19,23,42–44], and yet these features have curiously been little used in current WHO-led transmission interruption assessment survey designs[21,41]. This lacuna also includes the current incomplete information on the sensitivity and specificity of the various diagnostic tools used or proposed for measuring filarial infections[45]. These deficiencies mean that confidence in any predictions that parasite transmission interruption has occurred made by such frameworks is unlikely to be high. This supports the overriding need for not only resolving these infection measurement issues but also for evaluating the use of new predictive frameworks, such as the present PFFI tool, which can facilitate more informed decision-making via combining the effects of parasite extinction dynamics with diagnostic tool statistics effectively.

The application of our approach to programmatic disease-surveillance data tracking changes in human infection prevalences resulting from interventions against the two diseases investigated in this study (viz. onchocerciasis in Uganda and LF in Nigeria) has provided a first demonstration of how a model-based surveillance tool may be used to reliably establish as well as validate any reported infection-free status of a community. The results have highlighted two major benefits in this regard. First, if positive, albeit low-valued, infection breakpoints occur for a parasitic system, then they show that it is not necessary to wait until zero infection levels are reached before infection freedom evaluations can be attempted. This is a departure from previous applications of the disease freedom algorithms for making such assessments in the livestock and pest management settings[8,9,23], where zero infection sequence data are often used in making such calculations. The second benefit relates to its use for validating reported infection-free status based on the use of previously set or existing endpoint criteria. The chief result here pertains to our analysis of the data from the Mt. Elgon sites (Table 3), which show that primarily due to the infection threshold levels that apply to a setting, and the sample sizes and diagnostic tools used to measure infection levels, declarations of attainment of parasite elimination may well be supported significantly earlier than was actually done in a setting based on declines in infection prevalence alone. In addition, our analysis of sample sizes on the expected outcomes from the inconclusive sites observed in both the Madi Mid North focus and in Mt. Elgon (Table 5) point to the possibility that infection freedom may have been attained even in these sites if deficiencies in sample sizes used for carrying out the transmission interruption assessments were addressed adequately. These are clearly findings with major implications for the efficient design and surveillance of parasite elimination programmes. Getting such decisions right is also of major economic value as timely stoppings of control will lead to cost savings not only in terms of reducing unnecessary prolonging of interventions in any given area, but also via facilitating the re-allocation of resources from areas in which transmission is evaluated to be reliably interrupted to areas where the infection is shown to be potentially still present. Indeed, a coordinated assessment framework that combines the outcomes of data-driven parasite endpoint modelling and survey-based proof-of-freedom metrics may offer a tool for identifying the optimal stopping threshold in a setting by considering the uncertainty in infection freedom predictions, stakeholder preferences, and the potential monetary costs associated with surveillance and re-application of control if parasite freedom is wrongly declared[46].

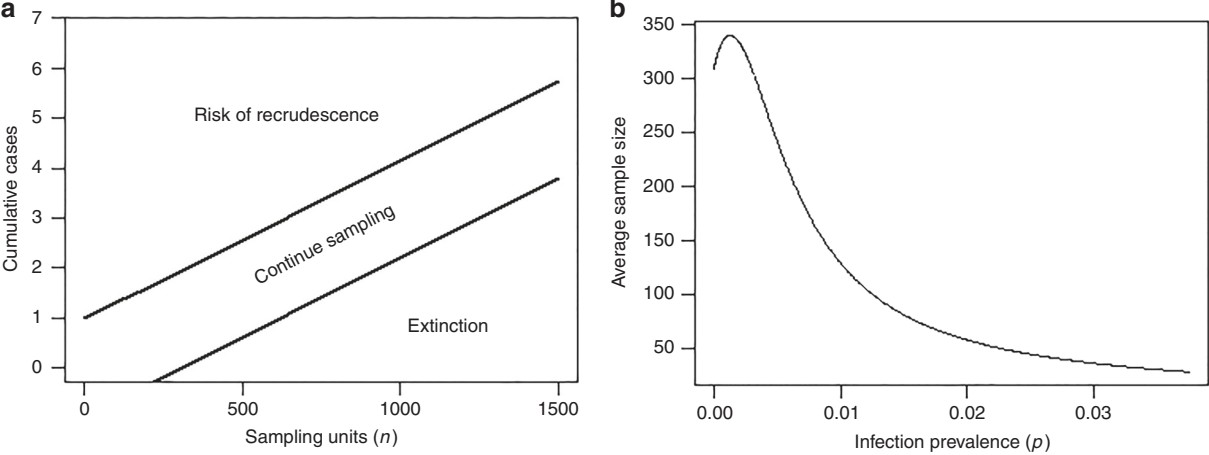

**Fig. 4** Sequential entomological sampling for classification. **a** Stop lines corresponding to a Wald's sequence probability ratio test (SPRT)[71] sampling plan for classification based on entomological infection thresholds, as measured by simple random sampling of vectors. Results for a scenario with $p_0 = 0.00005$ ($= 95\%$ EPT L3 prevalence threshold) and $p_1 = 0.0001$ ($= 5\%$ EPT threshold), $\alpha = 0.05$, $\beta = 0.10$ are presented. The cumulative total number of sampling units (mosquitoes or black flies) assessed as infected (say using dissections) is denoted on the y axis. n is the cumulative total number of sampling units assessed. Sampling units are observed sequentially and the cumulative total of those assessed as infected is noted. The criteria for cessation of sampling are characterized by the two parallel straight lines (the stop lines) shown. The lower stop line corresponds to the critical infection threshold (e.g., the 95% EPT L3 threshold prevalence), whereas the upper line denotes the emergence threshold (e.g., the 5% EPT L3 breakpoint prevalence value). If the observed data fall below the lower stop line following cumulative sampling bouts, the appropriate decision is to cease sampling and accept the null hypothesis (i.e., $p \leq p_{\text{CRIT}(= 95\text{EPT})}$). If the observed data fall between the two stop lines, sampling continues. If the data fall above the upper stop line, then the appropriate decision is to accept the alternate hypothesis (interpreted as $p \geq p_{\text{CRIT}}$). **b** Average Sampling number (ASN) curve for the sampling plan described above, showing that sample sizes per sampling bout will be maximal (approximately $= 350$ randomly sampled vectors) close to the critical prevalence adopted as a threshold value (i.e., $p = 0.00005$ in this case) and much smaller when p is substantially below or above $p_{\text{CRIT}}$

The present data analysis has also allowed appraisal of the next stage of critical work that must be resolved if the developed tool is to be deployed effectively in the field. First, with respect to our PFFI tool, we indicate that a major need is to validate whether estimating high infection freedom probabilities using human infection surveillance data and model-estimated design prevalences do indeed signify that parasite transmission has been broken in a site (e.g., > 95% confidence of freedom level (Tables 3, 4)). We have previously shown from analyses carried out in Papua New Guinea that one sensitive test to validate predictions of transmission interruption based on human infection thresholds is to assess whether crossing below these thresholds will lead to the absence of infection in vector populations[28]. However, one difficulty with assessing the crossing of the corresponding very low larval infection breakpoint prevalence values[6,47] or zero infection in vector populations is the requirement of significantly large sample sizes to carry out this exercise[48–50]. We suggest that the application of the sequential sampling approach for classifying whether an observed arthropod infection incidence is significantly below or above a critical intensity or threshold, however, may show particular promise for effectively resolving this practical problem in the field[49–52]. In sequential sampling, sampling efforts are carried out in sequence until a predefined critical threshold of pests or infected vector numbers, p, are either reached (so triggering implementation of control) or is detected to be below p (so enabling stopping of control). Figure 4 illustrates this method for a scenario for which the infective L3 larval threshold prevalence in a setting is 0.005%. The graph to the right of the panel indicates that to detect this threshold an average sample of at least approximately 350 mosquitoes would be needed. It is noteworthy that the equivalent sample size using the fixed n sampling approach (for the binomial distribution) for the same purpose would require up to 1200 mosquitoes per sampling bout[50,53]. The figure on the left shows that if 400 mosquitoes were sampled, say, but one infected

mosquito was found in a first bout, sampling should be continued, i.e., another bout of sampling should be conducted. However, suppose that in the second bout, zero infected mosquitoes are found, then the cumulative number of infected mosquitoes would still remain at 1. Now, reading along the y (cumulative positive cases) and x axes (observe that the cumulative sample size along the x axis is now 800) of the graph, we can use the stop lines (the lower of which represents the 0.005% threshold and the upper a 0.01% threshold[49,54] to determine at 95% confidence if the cumulative number of infected mosquitoes from a sequence of two bouts lies below the lower stop line or not. If it lies below the stop line, then sampling can cease and one can confidently predict (here at the 95% confidence level) that transmission has been broken (in the sense that infection in the sample of 800 mosquitoes from consecutive sampling is below the L3 prevalence threshold of 0.005%). These results show that coupling a sequential vector sampling framework to PFFI assessments based on human infection data in a setting could allow validation of the PFFI predictions of transmission interruption in a given site, including determining whether once infection thresholds are crossed, parasitic infection will decline to zero. Such an analysis will also permit evaluations of which indicator of infection might be most sensitive for supporting estimations of infection freedom. Empirical studies are now required for conducting these studies especially given that we are rapidly nearing the 2020 deadline set by the WHO 2012 Roadmap on NTDs for achieving the global elimination of these diseases[55].

A second future need is the extension of our method to account for serial surveillance data in order to develop plans for assessing cumulative evidence for infection freedom. Currently, we have considered each data point to be temporally independent, a simplification which is clearly less than optimal when dealing with the analysis of longitudinal surveillance data. Such plans must also consider the fact that populations are not closed and the effect that importation of infection will have on freedom

probabilities. We are investigating the application of Bayesian logic involving the specification of a prior probability and infection importation probability at each time step of calculation, as developed for the case of negative surveys[56,57], as a means to address the problem in the next stage of our work.

A third important need concerns how we might apply our prediction framework to support the effective making of area-wide freedom declarations. Although two-stage sampling strategies could be used to account for village-level clustering of infections, a more natural way forward, might be to couple maps of infection with the PFFI utility to carry out area-wide proof of freedom calculations based on the relative spatial distribution of infection risk over an area of interest[29,56]. Such a spatially explicit tool could also be used to determine the optimal spatial configuration of surveys and forecast the effort and costs necessary to declare success.

Although a generalized filarial transmission model was extended to simulate the specifics of LF and onchocerciasis transmission in this study based on previously established population models[6,15,24–26,58–60] (see Methods), it is important to note that, as for any model, new knowledge that would lead to refinements of the transmission processes incorporated in current models would result in changes to the breakpoint values used as design prevalences in the present PFFI tool. In particular, we highlight the important need to obtain better information on the various positive density-dependent factors that govern breakpoint thresholds in filarial infection, eg. functional form and operation of host immunosuppressive responses, co-variation in Mf distributions between host and vector populations, and worm death rate functions (see Methods)[15,61,62], if better numerical values of worm breakpoints are to be derived. These refinements would impact the infection freedom probabilities calculated here to a larger or smaller degree, further stressing the important need for continual efforts to update complex models and validate their predictions as new information regarding these processes arrive.

On a final note, we highlight that although our modelling approach was developed for supporting decision-making for the two diseases studied here, the developed framework is flexible and can be easily made specific for other parasitic diseases and management goals. This will be the case whether or not mathematical models that allow estimation of breakpoint transmission thresholds exist for these diseases, as in the absence of such predictions traditional FFI calculations based on zero survey data could still be used[38]. We are currently evaluating such enhancements, particularly with regard to the spatio-temporal and economic extensions required for (1) facilitating effective area-wide disease freedom assessments and (2) optimizing economically sound infection thresholds applicable to different settings, in our laboratory.

## Methods

**Calculating infection freedom probability using models.** Proving that a population is free from a disease or infection is difficult, if not impossible, due to the practical challenges of testing every individual and the limitations of diagnostic tools. The use of structured surveys and analyses to demonstrate freedom from infection must therefore rely on demonstrating a high probability that a population is free from infection given features of sampling and the diagnostic tools used to assess host infection status. Furthermore, it is not necessary to reduce infection levels to zero before stopping interventions, as according to epidemic theory, there exists a threshold infection prevalence below which transmission is no longer sustainable and the system will tend towards a state of zero infection without further intervention[15]. This theoretical threshold is termed as the infection breakpoint and can represent the target infection level to cross below in order to stop interventions. For surveys to determine infection freedom, this prevalence is the infection level a survey is designed to detect with a known probability and is termed the design prevalence[8,9,43].

**Probability calculations.** Central to the freedom from infection analysis is the estimation of whether the probabilities of the null ( = infection is present at a level

equal to or greater than the design prevalence) and alternative ( = infection is present at a level less than the design prevalence) hypotheses differ significantly given the observations of the survey and the characteristics of the diagnostic test. The assessment of the null hypothesis ($P_0$) is carried out by quantifying the probability of observing $x$ or fewer positive samples ($T^+$) in a sample of size $n$ with disease prevalence equal to the design prevalence ($p = p_d$), whereas the alternative hypothesis ($P_a$) is evaluated via estimating the probability of observing $x$ or more positive samples in a population free of disease ($p = 0$). In general, using the frequentist approach, the probability of getting $x$ positive individuals, $P(T^+ = x)$, in a sample of size $n$ with an infection prevalence equal to $p$ is given by a modified binomial distribution where the diagnostic test is considered to be imperfect with a known sensitivity (Se) and specificity (Sp). Combining probabilities of observing true and false positives given Se and Sp of a test, we obtain[8]:

$$P(T^+ = x) = \binom{n}{x} [p\text{Se} + (1-p)(1-\text{Sp})]^x$$

$$[p(1-\text{Se}) + (1-p)\text{Sp}]^{(n-x)}$$

(1)

This formula assumes an infinite population and calculates the probability of sampling $x$ positives in $n$ samples with replacement. As many of the populations relevant to disease monitoring sites are relatively small ( < 10,000 individuals in a study area), we adjust the calculation to be appropriate for a finite population where individual tests are not independent (probability of selecting $x$ positives in $n$ samples without replacement). To achieve this, a hypergeometric distribution is similarly modified[8]:

$$P(T^+ = x) = \sum_{y=0}^{d} \frac{\binom{d}{y}\binom{N-d}{n-y}}{\binom{N}{n}}$$

$$\times \sum_{j=0}^{\min(x,y)} \binom{y}{j}\text{Se}^j(1-\text{Se})^{y-j}\binom{n-y}{x-j}(1-\text{Sp})^{x-j}\text{Sp}^{n-x-y+j}$$

(2)

where $d$ is the number of diseased individuals in a population of size $N$ ($d = p^\star N$), $y$ is the number of diseased individuals in a sample of size $n$, $x$ is the number of individuals that test positive in the sample, and $j$ is the number of true positives in the sample and $x-j$ is the number of false positives. The probabilities of the null and alternative hypotheses given this distribution are calculated as:

$$P_0 = \sum_{i=0}^{x} P(T^+ = i) \text{where } p = p_d$$

$$P_a = \sum_{i=x}^{n} P(T^+ = i) \text{where } p = 0$$

(3)

In order to sufficiently substantiate the claim that a population is free from infection, the probability of the null hypothesis must be low and the probability of the alternative hypothesis must be high, according to the desired rates of Type I ($\alpha$) and Type II ($\beta$) error. For instance, if $\alpha$ and $\beta$ are both set to 0.05, then we require $P_0 \leq 0.05$ and $P_a \geq 0.95$, in order to characterize a population as free from infection at the desired confidence level. The probability of being free from disease is therefore defined as:

$$P(\text{free}) = 1 - P_0$$

(4)

In the event that the probabilities of both the null and alternative hypotheses are high, there is insufficient evidence to conclude whether or not the population is free from infection due to a small sample size[8,11,23].

In order to conduct a PFFI analysis, the following details about the survey, diagnostic tool, and analysis specifications are thus required as data inputs: $N$ = total population, $n$ = number of individuals sampled, $x$ = number of individuals who test positive, $p_d$ = parasite transmission breakpoint (design prevalence), Se = diagnostic sensitivity, Sp = diagnostic specificity, $\alpha$ = rate of Type I error, and $\beta$ = rate of Type II error. In the analyses conducted here, the inputs included the LF and onchocerciasis Mf survey data and breakpoint thresholds ( = design prevalence) (Tables 1, 2), as well as the following specifications: $\alpha = 0.05$ and $\beta = 0.05$, and Mf detection performance statistics for skin snip and blood smear examination given by Se = 0.95, Sp = 0.95. It is noteworthy that the values chosen for Se and Sp are somewhat arbitrary due to a lack of consensus on the performance of these diagnostic tools[31–37] and are applied here with the intention of allowing a reasonable first demonstration of the use of the present tool. We provide a R function that can be used to carry out these calculations in the Supplementary Note 1.

**The filarial transmission dynamics model.** In this study, we employ a generalized immigration-death model describing the transmission of filarial parasites in both

human and vector hosts to carry out the LF and onchocerciasis modelling work. We have previously described this model in detail for describing the transmission and elimination of LF[6,15,25,63] and we present the structurally similar onchocerciasis model here for the first time. Briefly, this model simulates the population level dynamics of various life stages of a filarial parasite in both human and vector hosts via the use of coupled partial differential and ordinary differential equations for describing changes in the pre-patent worm burden per human host ($P(a,t)$), adult worm burden per human host ($W(a,t)$), the Mf level in the human host ($M(a,t)$), the average number of infective L3 larval stages per vector ($L$), and a measure of immunity ($I(a,t)$) developed by human hosts against L3 larvae. The state equations describing the model are as follows

$$
\begin{aligned}
\frac{\partial P}{\partial t} + \frac{\partial P}{\partial a} &= \Phi L^* F_1(I(a,t)) F_2(W_T(a,t)) \\
&\quad - \mu_w P(a,t) - \Phi L^* F_1(I(a,t-\tau)) F_2(W_T(a,t-\tau))\zeta \\
\frac{\partial W}{\partial t} + \frac{\partial W}{\partial a} &= \Phi L^* F_1(I(a,t-\tau)) F_2(W_T(a,t-\tau))\zeta - \mu_w W(a,t) \\
\frac{\partial M}{\partial t} + \frac{\partial M}{\partial a} &= F_3(W_T(a,t)) - \gamma M(a,t) \\
\frac{\partial I}{\partial t} + \frac{\partial I}{\partial a} &= W_T(a,t) - \delta I(a,t) \\
L^* &= F_4(W_T(a,t))
\end{aligned}
\tag{5}
$$

The state variables and parameters of this model for both LF and onchocerciasis are provided in Supplementary Tables 2 and 3. Here we note that the term, $F_x$, denotes the density-dependent functional forms that govern the transitions or development rates of different parasite states in the life cycle. It is noteworthy that some functions are dependent on the total worm load where $W_T(a,t) = W(a,t) + P(a,t)$, whereas others depend on larval states ($L^*$) and host immunity ($I$). The functional forms for the LF and onchocerciasis models are similar with two primary differences: (1) in the onchocerciasis model, the larval death rate and excess vector mortality due to infection is considered in $F_4$, while these terms are ignored in the LF model, and (2) the vector biting rate (a model parameter in the function $\Phi$) considers the number of bites per black fly to be equal to the human blood index divided by the gonotrophic cycle while the number of bites per mosquito is a standalone parameter in the LF model. The mathematical representation of adult *Onchocerca volvulus* worm mortality has recently been a topic of discussion[64]; here we use a constant mortality rate following the example of Basáñez and Boussinesq[58] and Filipe et al.[59]. Given that previous modelling studies have underscored the potential for the involvement of either or both acquired immunity and immunosuppression in filarial infections[59,60], we include both these types of immunity in our generic filarial models; however, it is worth noting that we do not make any a priori assumptions concerning their occurrence in the present study sites but instead use the Mf prevalence data to determine the operation of each or both in each site. All other parameters appear in both models, with each parameter's value estimated using the corresponding field data for each disease (full details provided in Supplementary Tables 2, 3).

**Bayesian Melding fitting of the filarial models to data.** Our data-driven modelling approach is essentially based on integrating field data into the filarial models in order to estimate models that can reliably capture the local transmission dynamics in a particular setting. We employed a data assimilation-based inverse modelling framework founded on the Bayesian Melding (BM) procedure to undertake such model estimations using local human infection (Mf prevalence) and vector-related transmission (ABR) data[6,24–26] as follows. Briefly, we begin by using known or uniformly assigned (if values are unknown) ranges in parameter values to generate distributions of parameter priors. We then randomly sample with replacement from these prior distributions to generate 200,000 parameter vectors, which are then run using the ABR values, if given, for a site to generate model outputs. In the event that baseline ABR information is not available, as is the case for the sites modelled here, the ABR is sampled as one of the model parameters whose prior uniform range is bounded to be reasonable for the given geographical setting. The model outputs are then evaluated against the observed age-stratified Mf prevalences by calculating the binomial log-likelihoods of each parameter vector for the data. In the resampling step of the BM method, a Sampling-Importance-Resampling algorithm is used to perform 500 draws with replacement from the pool of parameter vectors generated in the step above, with probabilities proportional to their relative log-likelihood values. This step generates the most likely parameter vectors or models describing the data. These resampled parameter vectors are then used to generate distributions of variables of interest (e.g., age-prevalence curves, worm breakpoints and infection trajectories following treatments)[6,25,26]. It is noteworthy that the use of this Bayesian data assimilation procedure means that the values of none of the model parameters are fixed in advance; rather values are derived for a site by allowing the corresponding data to select the best-fit posterior values from initially set prior distributions for each parameter.

**Hindcasting baseline conditions from post-intervention data.** In some areas, no baseline Mf prevalence or ABR information might be available and the first instance

of data may relate to a time after which treatments have already been administered. In these cases, we used the model to hindcast baseline conditions by relying instead on post-intervention data and details of the treatment programme. We initialize the model by selecting 200,000 parameter vectors, which produce plausible baseline Mf prevalence and ABR conditions (for instance, 5–60% LF Mf prevalence in Nigeria with ABR values 500–5000). The ensemble of baseline simulations is then modelled forward in time applying the appropriate interventions (see Supplementary Methods for details regarding the modelling of MDA in these sites). These outputs are then resampled using a pass/fail filter[65,66] to select the 500 models which most closely follow the trends in Mf prevalence as given by post-intervention data. This resampled subset of models is taken as the posterior sample of parameter vectors to reproduce or hindcast the baseline infection/ABR values applicable in a given site[26].

**Calculations of site-specific transmission thresholds.** A numerical stability analysis approach was applied to each of the best-fitting parameter vectors in order to calculate the TBR, as well as the distribution of Mf prevalence breakpoints expected in a community[6,15,24–26]. To calculate the TBR for each parameter vector using this method, we begin by keeping all model parameters constant and progressively decreasing the average number of black flies (or mosquitoes) per human, $m$, from its original value to a threshold value below which the model always converges to the zero Mf prevalence. The product of the number of bites per fly per month, $\beta$, and this newly found $m$ value is termed as the TBR. Given a particular biting rate (either ABR or TBR), the model will thus settle to either a zero or non-zero Mf prevalence depending on the initial value of $L^*$. Therefore, by starting with a very low value of $L^*$ and progressively increasing it in small step sizes we estimate the minimum $L^*$ below which the model predicts zero Mf prevalence and above which the system progresses to a positive endemic infection state[6,15,24–26]. The corresponding Mf prevalence at this threshold $L^*$ value is termed as the worm/Mf breakpoint. This process is repeated for both the site-specific ABR and TBR as required.

The distribution of Mf breakpoints at a particular biting rate in a site can be described by an inverse ECDF. We used this function, in conjunction with exceedance calculations[25], to quantify the values of Mf breakpoint prevalence thresholds reflecting various elimination probabilities in a site. Here we used the Mf threshold value corresponding to 95% elimination probability (the 95% EPT) as the desired threshold value for serving as the design prevalence in all the PFFI calculations reported.

**The integrated PFFI tool.** Figure 2 illustrates the components of our integrated proof of parasite freedom tool that allows the coupling of infection survey data from sentinel sites undergoing interventions with filarial transmission model predictions of breakpoints, to facilitate the making of probabilistic freedom from infection calculations. As shown by the diagram, the process begins by assembly of surveillance data on both pre- as well as post-intervention infection prevalences from monitoring sites. In the case of pre-intervention infection data, both the vector biting intensity in the form of ABR and age-prevalences together with sample sizes are ideally collated for each site, although we have developed model-based procedures to estimate these data if only overall prevalence data are available[66]. In the second step, the filarial transmission models are fitted to the available or estimated pre-intervention survey data using BM as described above, and the set of best-fitting models for a site are then numerically evaluated to calculate infection breakpoint values (for all relevant indicators, including Mf, ICT, and L3 prevalence, as required) applicable to a particular site. In the third step, these breakpoint values are used as design prevalences in the PFFI calculator in conjunction with post-intervention infection prevalence survey data, to carry out calculations of the probability that parasite infection freedom has been achieved using the statistical methodology described above. Note, while we focus on site-specific PFFI calculations here, the method can also be extended across an entire spatial domain of interest (e.g., by the application of two-stage sampling approaches[9] discussed as a possible method for evaluating area-wide malaria freedom by Stessman et al.[38] recently, but see Discussion).

**Infection survey data.** We partnered with The Carter Center (TCC) to demonstrate the application of the integrated PFFI tool for supporting declarations of infection freedom in communities that have either received or are undergoing anti-filarial interventions by using actual infection survey data from the TCC-supported Onchocerciasis Elimination Programme in Uganda, as well as published and unpublished surveillance data from the Nigerian LF Elimination Programme[30]. Table 1 lists the onchocerciasis monitoring data evaluated in this exercise obtained from two transmission foci in the country, one (Mt. Elgon focus) in which parasite transmission by *Simulium neavei*[67] has been declared to be interrupted by the programme, and the other (Madi Mid North focus) in which transmission by *Simulium damnosum*[68] is considered to be currently ongoing. The transmission status of a focus is defined by the programme based on stopping criteria adopted by Uganda Ministry of Health, which requires the achievement of < 1% Mf prevalence to demonstrate the elimination of morbidity, an absence of infection in children younger than 10 years old and an absence of infected vectors for a period of 3 years (*S. neavei*) or achieving L3 prevalence < 0.05% in the case of *S. damnosum*, to confirm the interruption of transmission[67]. Data on Mf prevalence including sample sizes were available at various follow-up times for the sentinel sites in each

focus, but although baseline infection data were available, sample sizes were missing (Table 1). Random selection of subjects was followed at each infection evaluation point. With regards to interventions, annual MDA using Ivermectin was initiated and implemented in Mt. Elgon from 1994 to 2006, but treatments were switched to bi-annual application from 2007 until 2011. In addition, vector control using Abate was added to bi-annual MDA from the year 2007. All interventions in the focus were stopped in the year 2011. In the case of Madi Mid North, by contrast, annual MDA was initiated in the year 1995 and this remained the mainstay of the programme in that focus to the present day. MDA coverage information were also available. All Mf infection assessments were done using the skin snip method.

The corresponding surveillance data describing changes in LF Mf infection prevalence for sentinel villages that received > 8 years of annual MDA with Ivermection and Albendazole in two example Nigerian regions where transmission is primarily mediated by Anopheles mosquitoes[30] are shown in Table 2. Annual MDA began in different years in these villages, and data describing LF Mf prevalences at baseline and at various intervention follow-up times together with the applicable sample sizes were available for analysis. Nocturnal Mf assessments were done using microscopy inspection of 60 μl thick smear blood samples[30]. MDA coverages were also available for each site.

**Code availability**. The Matlab code for running the LF and onchocerciasis models used in this work is available at https://github.com/EdwinMichaelLab/PFFI. The R code used to calculate the probability of infection freedom at the modelled design prevalence or parasite transmission breakpoint value given different diagnostic tools and sample sizes used by surveys is given as a ready to run R function in Supplementary Note 1.

## Data availability

The authors declare that the data used in support of this study and its findings are available within the paper and in the references pertinent to each dataset, or are available from the authors upon request.

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

## Acknowledgements

We are grateful to the Vector Control Division, Ministry of Health, and the Onchocerciasis Elimination Programme of Uganda for the surveillance data on onchocerciasis used in the present analysis. We also thank the Nigerian LF Elimination Programme and staff of The Carter Center's Nigeria offices, whose hard work provided the published survey data for carrying out the PFFI calculations for LF reported in this manuscript. This work was made possible by the financial support provided by the National Institutes for Health, USA, on grant number: RO1AI123245.

## Author contributions

Conceived the paper: E.M. Conducted the analysis: M.E.S. and E.M. Wrote the first draft of the manuscript: E.M. and M.E.S. Contributed to the writing of the manuscript: all authors. Agree with the manuscript's results and conclusions: all authors.

## Additional information

**Competing interests:** The authors declare no competing interests.

