## [Peer Review File · Nature Communications]

Reviewers' comments:

Reviewer #1 (Remarks to the Author):

This manuscript presents a new computational framework for guiding decision making on stopping interventions for parasitic disease elimination programs. The authors have done a very good job in 1) demonstrating the utility of this tool for supporting intervention cessation decisions using empirical data for onchocerciasis and lymphatic filariasis control program in Uganda and Nigeria, 2) providing insights on how to potentially validate their model results in the field, and 3) discussing the limitation of their study.

I only have a couple of comments which I think should be addressed in the manuscript.

1) The authors should state if the existence of breakpoints, which is paramount to this method, has been empirically observed for any of this parasitic disease or if it remains only a theoretical concept.

2) If it is only a theoretical concept predicted by the model, but which may or may not occur in the field, a policy decision erroneously driven by the existence of a model-based breakpoint would likely result in disease reemergence. How should disease surveillance account for this eventuality?

3) Given the spatial heterogeneity in disease prevalence in both human and vector populations, the geographical unit for disease surveillance/sampling can affect the reliability of results as the presented methods assume homogeneous risk within the human and vector populations within the area of interest. This raises questions about the optimal grid size for disease sampling for estimating infection freedom threshold in a given area. The authors should discuss this issue.

Reviewer #2 (Remarks to the Author):

In this manuscript, Michael and co-authors present an interesting analysis using the concept of freedom from infection approach (from the veterinary field) and applying it to human filarial infection. My main concerns are not with the approach itself, but with the premises, assumptions and models on which the approach is predicated.

Major Comments:

1) Firstly, the authors use microfilarial prevalence thresholds that have been advocated by major control programmes as operational thresholds for cessation of mass treatment as their transmission breakpoints. Apart from the fact that the source of these values should be appropriately referenced (e.g. APOC 2010), other modelling studies (e.g. Stolk et al. 2015 Parasit Vectors; Walker et al. 2017 Epidemics) have shown that these operational thresholds (e.g. 1% microfilarial prevalence, taken here as the 'design prevalence'), are not applicable to all endemicities and epidemiological settings. Instead, they will depend on the

baseline (pre-intervention) prevalence and conditions for transmission. In areas of initially high prevalence and transmission intensity, the true transmission breakpoints (i.e. the values of parasite intensity and prevalence below which the parasite population would terminally decline) are likely to be much lower than the proposed operational thresholds. The authors acknowledge this in the Introduction and discuss it well in the Discussion section, but the results seem to indicate something different. In Table 1A, resulting values of Mf breakpoint (so-called design) prevalence are given for the onchocerciasis focus of Mt. Elgon in Uganda, where transmission has been interrupted by a combination of vector control (against *Simulium neavei*) and ivermectin treatment. They are lower than 1% at the threshold biting rate (TBR), but not very different between hyperendemic and mesoendemic areas. The TBR values are not given. Since the basic reproduction number must have been greater in areas of higher endemicity, the TBR must have been lower, but how the values of TBR vary once treatment is introduced is not stated. Interestingly, the Mf breakpoint prevalence values for the ongoing transmission focus of Madi Mid North decrease with initial endemicity, as expected. What are the annual biting rates (observed or inferred) in each locale? And what is the relationship between microfilarial prevalence and ABR? This is a crucial relationship for the probability of elimination (as shown by Walker et al. 2017). The other question is how has the negative binomial distribution (of microfilarial load? Of worm load?) been parameterised. In the Supplementary references, all the references seem to pertain to LF, and it is unclear to which parasite stage they refer to and how relevant these parameters are for onchocerciasis. Other authors have explored this in the context of onchocerciasis (Walker et al. 2017).

2) Second, the authors use fixed values for the diagnostic performance (sensitivity and specificity) of the tests (skin snip microscopy for onchocerciasis and blood sample microscopy for LF), regardless of how the sensitivity may vary with the intensity of infection. The assumed values do not seem motivated by literature; at least a reference is not given, and I believe that the specificity could be higher (the microfilariae of the different skin-dwelling or blood-dwelling filariae are identifiable), but the sensitivity will vary with microfilarial load. At least in onchocerciasis, there will be an aggregated distribution of microfilariae in the skin and it has been shown by Bottomley et al. (2016, Parasit Vectors) that sensitivity varies with infection intensity, time after the last treatment, assumed impact of treatment on female worm microfilarial production and number of snips taken. I doubt that the sensitivity of the diagnostic will remain 95% in near elimination settings. In Figure 2, it is not very clear how the diagnostic performance characteristics relate to sample size, but it is likely that specificity will be very high, whilst sensitivity will likely be a lot lower than the lower value of 80% displayed.

3) Last, but not least, the generic filariasis model that the authors use is a lymphatic filariasis model (originally developed by Norman and co-workers) presented elsewhere, and it is not applicable to onchocerciasis. The reasons for this are several:

3.i) The assumption of a constant mortality rate of adult worms (parameter μ sub W) implies an exponential distribution of adult worm survival times that is at odds with observations on microfilarial prevalence and intensity made after interruption of transmission under large-scale vector control (Plaisier et al. 1991 Acta Trop). These

observations have led to modifications of onchocerciasis models by other authors (Walker et al. 2017 Epidemics). In onchocerciasis, treatment with ivermectin is not strongly and fast acting macrofilaricidal, so assumptions on the survival times of the adult worms do matter (they matter a lot less for LF, as the combination of ivermectin and albendazole seems to have a more pronounced effect on the survival of adult *Wuchereria bancrofti*).

3.ii) The assumptions that immunity depends on mean adult worm load (equation for state variable I), and the operation of acquired immunity dependent on adult worm load (that enters into the equation for pre-patent infection, P) are not substantiated. In the Supplementary material, the authors tabulate parameters for immunosuppression affecting parasite establishment within humans (a hypothesis presented by Duerr et al. *Trans R Soc Trop Med Hyg*). However, other authors have argued that the different shapes of microfilarial intensity (not prevalence, which is a lot less informative given the nonlinear relationship between microfilarial prevalence and intensity) versus host age are due to differences in exposure (with age and among the sexes), without the need of invoking immunosuppression (Filipe et al. 2005, *PNAS*).

3.iii) As I understand the parameters used from the Supplementary table, the authors have plucked a number of parameter ranges from the modelling literature, i.e. from a number of modelling studies that have tested different hypotheses about parasite population regulation within the human host, where we have most uncertainty. By doing so, they are implicitly assuming that the parameters are uncorrelated among themselves and independent of structural assumptions. This is not the case. In fact, and for instance, the proportion of infective larvae that establish within the humans is taken from models presented by Basáñez et al. (2002, *Am J Trop Med Hyg*) and Filipe et al. (2005, *PNAS*), but these parameters were estimated from fitting the models to data purposely ignoring immunosuppression. In turn, the parameters of immunosuppression were estimated by Duerr et al. ignoring the role of exposure. Hence, if the model the authors are using has both sets of mechanisms, exposure and immunosuppression, surely the parameters would have to be re-estimated from data and the hypotheses of either mechanism or both tested as structural and parametric assumptions given the importance of the exposure function. I don't think that using an 'off-the-shelf' LF model and using parameter sets that have been estimated with very different onchocerciasis models is the correct approach.

3.iv) The F functional forms for density-dependent processes are given in the Supplementary tables. In F3, for instance, one can envisage a combination of mating probability (positive density dependence) and density-dependent worm fecundity (negative density dependence), but this is not very clear and has different dynamic implications. In fact, another paper by Duerr et al. (*Int J Parasitol*) invokes density-dependent adult worm fecundity in onchocerciasis savannah settings. The existence of a breakpoint parasite density is dependent on the operation of positive density dependence. The authors state that function F4 contains larval mortality in onchocerciasis, but do not explain how the functional form may differ between *Simulium damnosum* and *S. neavei*. In the former, for instance, the functional form of microfilarial uptake (limitation) and fly survival (decreasing with microfilarial intake) are important as described in other papers (reviewed by Basáñez et al. 2009, *Adv Parasitol*), but it is not clear what exactly goes into F4 for each simuliid

species and how this is parameterised, even when referring to the Supplementary tables.

3.v) In Africa, and for LF, vectors such as Anopheles species do have a cibarial armature (as opposed to Culex), but there are no onchocerciasis vectors with cibarial armature in Africa. This is not clearly stated, so it is difficult to understand which functional form goes with what exactly. This makes it difficult for other researchers to reproduce the work.

In conclusion, I think the authors present an interesting approach but should focus solely on lymphatic filariasis, for which they have developed and parameterised a model that is specifically tailored to this disease.

Minor comments:

4) I found reading the Abstract that it was not very informative of what precisely the paper is about. Using clearer, more concise language would be helpful to the reader. Some sentences are very long and difficult to read, or lack punctuation, and would benefit from rephrasing and dividing into smaller, more digestible sentences. For instance: "Coupling infection survey-based metrics with parasite transmission dynamics model predictions of extinction thresholds can maximise the utility of information related to the potential infection status of a population to support objective calculations of the probability of achieving infection freedom."

And at the beginning of the Results section: "Figure 1 provides an illustration of the ability of our modelling approach for not only learning best-fitting filarial models for observed or derived age-stratified Mf prevalence data in different community settings, but also for estimating Mf transmission thresholds for use as design prevalences based on the discovered best-fitting models or parameter vectors from each site."

"Results are shown for two of the Onchocerciasis and one of the LF study sites in the figure, with the full set of model fits and the corresponding 95% Mf elimination thresholds (95% EPTs) at either the Annual Biting Rate (ABR) or the Threshold Biting Rate (TBR) values derived using the inverse cumulative density function (CDF) approach in all the other present study sites (depending on whether a particular location has implemented vector control measures in addition to Mass Drug Administration (MDA)) given in Supplementary Figures S1 and S2."

These are only examples, but I found the language to be very verbose; sentences should be shorter, more concise and made clearer to the reader.

5) The definitions and goals of eradication and elimination are different and should not be used interchangeably.

By the same token, the definitions of infection and disease are different and should not be used interchangeably. Elimination of disease (presumably morbidity) is very different from elimination of infection.

- 6) Some references have been left as EndNotes (e.g. {6 Sunish, 2002 #50}).
- 7) Spelling of program and programme is inconsistent; it should be the latter.
- 8) Maximum acceptable disease prevalence or design prevalence should be defined from the outset.
- 9) Spell out all abbreviations/acronyms when first used (e.g. WHO, EPT).
- 10) There is a number of typos. Data is a plural noun.
- 11) References need formatting according to journal style.

Response to Reviewers

Reviewer #1 (Remarks to the Author):

This manuscript presents a new computational framework for guiding decision making on stopping interventions for parasitic disease elimination programs. The authors have done a very good job in 1) demonstrating the utility of this tool for supporting intervention cessation decisions using empirical data for onchocerciasis and lymphatic filariasis control program in Uganda and Nigeria, 2) providing insights on how to potentially validate their model results in the field, and 3) discussing the limitation of their study.

I only have a couple of comments which I think should be addressed in the manuscript.

1) The authors should state if the existence of breakpoints, which is paramount to this method, has been empirically observed for any of this parasitic disease or if it remains only a theoretical concept.

We thank the reviewer for raising this important point. We have published preliminary evidence for the existence of thresholds in lymphatic filariasis in Reimer et al. 2013 (NEJM). This work shows for five Papua New Guinea sites that 1) although five rounds of MDA reduced mf prevalence to as low as 1.7%, it did not reduce mf prevalence below the site-specific mf breakpoints estimated in Gambhir et al. 2010 (BMC Biology) and, as a result, transmission was still ongoing 10 years later, and 2) that reducing the biting rate below the model-predicted biting rate thresholds with high probability through the use of bednets coincides with finding no detectable transmission as measured by number of infective larvae/person/year.

We have discussed the concept of non-zero transmission breakpoints in the introduction and discussion sections, and have added a sentence to clarify that this is an empirically observed phenomenon with this later study as a reference on lines 147-152.

2) If it is only a theoretical concept predicted by the model, but which may or may not occur in the field, a policy decision erroneously driven by the existence of a model-based breakpoint would likely result in disease reemergence. How should disease surveillance account for this eventuality?

As stated in the response to point 1, these breakpoints are not purely theoretical and there exists empirical evidence for the non-zero transmission thresholds predicted by mathematical models. The reviewer raises an important issue with regards to reemergence in the case that thresholds are not reached prior to stopping interventions. In fact, reemergence is being observed in some areas, suggesting that the thresholds defined by the WHO do not apply everywhere. In response to this, our model accounts for heterogeneity in transmission thresholds and identifies location-specific breakpoints which are often lower than those proposed by the WHO (Gambhir et al. 2010, Singh et al. 2013, Singh and Michael 2015, Michael and Singh 2016). Additional validation studies are still needed to empirically test the model-predicted breakpoints, but it is clear that heterogeneous dynamics need to be considered and that post-treatment surveillance is a critical aspect in these in control programs.

The threat of reemergence and the urgent need for additional breakpoint validation studies have been discussed in the discussion section of the manuscript (see lines 253-264, 318-322, 349-353, 391-397).

3) Given the spatial heterogeneity in disease prevalence in both human and vector populations, the geographical unit for disease surveillance/sampling can affect the reliability of results as the presented

methods assume homogeneous risk within the human and vector populations within the area of interest. This raises questions about the optimal grid size for disease sampling for estimating infection freedom threshold in a given area. The authors should discuss this issue.

We fully agree with the reviewer that geographical heterogeneity is an important consideration in control efforts for these two diseases, and that additional work is required to identify the optimal spatial scale of surveys. As a first proof of principle, this study focuses on a single stage analysis at the village level in part due to the availability of data and to use the unit size currently relevant to control programs. We agree that the future work should consider spatial analyses and two-stage survey methods.

We have highlighted this research need in the penultimate two paragraphs of the discussion section of the manuscript, including pointing to possible extensions to two-stage sampling methods and/or the coupling of freedom calculations to maps to address this issue. This is a current focus of research in our group.

Reviewer #2 (Remarks to the Author):

In this manuscript, Michael and co-authors present an interesting analysis using the concept of freedom from infection approach (from the veterinary field) and applying it to human filarial infection. My main concerns are not with the approach itself, but with the premises, assumptions and models on which the approach is predicated.

Major Comments:

1) Firstly, the authors use microfilarial prevalence thresholds that have been advocated by major control programmes as operational thresholds for cessation of mass treatment as their transmission breakpoints. Apart from the fact that the source of these values should be appropriately referenced (e.g. APOC 2010), other modelling studies (e.g. Stolk et al. 2015 Parasit Vectors; Walker et al. 2017 Epidemics) have shown that these operational thresholds (e.g. 1% microfilarial prevalence, taken here as the 'design prevalence'), are not applicable to all endemicities and epidemiological settings. Instead, they will depend on the baseline (pre-intervention) prevalence and conditions for transmission. In areas of initially high prevalence and transmission intensity, the true transmission breakpoints (i.e. the values of parasite intensity and prevalence below which the parasite population would terminally decline) are likely to be much lower than the proposed operational thresholds.

Although we discuss the operational thresholds suggested by major control programs, we do not use these values for the design prevalence in our analysis. We apologize that this was misunderstood by the reviewer, indeed our Bayesian data-driven modelling approach described under Methods addresses this question explicitly by using data to discover locally applicable models and hence breakpoint thresholds. We did also point out based on results from our previous studies (lines 103 – 106) that these thresholds will vary significantly from site-to-site in our calculations. We have noted how applying our models to data can allow allows us to consider variability in transmission thresholds stemming from ecological heterogeneity (lines 115-120). As pointed out by the reviewer, these thresholds are indeed much lower than the suggested operational thresholds (see Table 1), which we had also summarized in line 174. We have also added a reference for the suggested APOC reference in addition to the WHO publication referenced for the LF threshold on line 95. We have also added the other references suggested by this Reviewer.

The authors acknowledge this in the Introduction and discuss it well in the Discussion section, but the results seem to indicate something different. In Table 1A, resulting values of Mf breakpoint (so-called design) prevalence are given for the onchocerciasis focus of Mt. Elgon in Uganda, where transmission has been interrupted by a combination of vector control (against *Simulium neavei*) and ivermectin treatment. They are lower than 1% at the threshold biting rate (TBR), but not very different between hyperendemic and mesoendemic areas. The TBR values are not given. Since the basic reproduction number must have been greater in areas of higher endemicity, the TBR must have been lower, but how the values of TBR vary once treatment is introduced is not stated. Interestingly, the Mf breakpoint prevalence values for the ongoing transmission focus of Madi Mid North decrease with initial endemicity, as expected.

What are the annual biting rates (observed or inferred) in each locale? And what is the relationship between microfilarial prevalence and ABR? This is a crucial relationship for the probability of elimination (as shown by Walker et al. 2017).

The ABR values for the sites modelled in this work were estimated because no data were available. We have edited the text on lines 492-494 to clarify that this is the case for the sites modelled here. The ABR is treated as a parameter to be sampled, with minimum and maximum values chosen to reflect reasonable values for the geographical and ecological characteristics of the site (see line 493) and in discussion of Bayesian Melding approach in response to point 3.iii below). The ABR is therefore informed by the mf prevalence data used to fit the model but is represented as a distribution per the Bayesian approach rather than a point estimate. The estimated ABR, along with the mf prevalence and fitted model parameters, are used to calculate the site-specific thresholds biting rates and breakpoints. As the reviewer points out, the breakpoints and thresholds biting rates will vary based on endemicity, but this is not the only characteristic playing a role as we have shown in our previous papers (Singh & Michael 2015 *Parasites & Vectors*, Michael & Singh 2016 *BMC Medicine*). Other factors include the baseline biting rate (which here is unknown), parasite aggregation, and immunity. These factors, when acting together, create nonlinear dynamics which make drawing conclusions about trends difficult with the relatively small sample of sites investigated in this work. Rather as we have stressed through this paper we allow the site-specific data to inform model estimates, i.e. we do not set up any *a priori* ideal expectation regarding any mf prevalence and ABR relationship for any of our sites. As pointed out by theoreticians working with heterogeneous spatial systems, if spatial nonstationary occurs in any pattern (mf prevalence) – process (model parameters) relationship, i.e. if such pattern-process relationship changes significantly across a heterogeneous spatial domain, then using any idealized global functional relationship will increase bias and hence increase the uncertainty in model predictions (Constanza & Voinov 2004, Cushman 2010, Cushman et al. 2010, Beven 2009).

The other question is how has the negative binomial distribution (of microfilarial load? Of worm load?) been parameterised. In the Supplementary references, all the references seem to pertain to LF, and it is unclear to which parasite stage they refer to and how relevant these parameters are for onchocerciasis. Other authors have explored this in the context of onchocerciasis (Walker et al. 2017).

The negative binomial distribution describes the mf and worm loads (k enters the equation for worm mating probability as well as vector mf uptake, same parameters k_0 and k_{Lin} are assumed for both mf and worm burdens $k=k_0+k_{Lin}*(W \text{ or } M)$). We apologize that our parameterization method has not been made clear to the reviewer and we have added further clarification in the main text and supplementary information. In our Bayesian Melding approach, the use of uniform priors presented in Table S1 reflect

the uncertainty in the parameter values and serve as bounds for what would be reasonable based on empirical data. The posterior parameter values ultimately used in the modelling exercises are estimated from data (see discussion of BM in response to point 3.iii below and lines 483 – 505 in the main text). In general, helminths are expected to be overdispersed, which has been shown to be true for onchocerciasis in Basáñez and Boussinesq 1999 (Phil. Trans. R. Soc. Lond. B), Filipe et al. 2005 (PNAS), Bottomley et al. 2016 (Parasites & Vectors) and for LF (Michael et al. 2001 Parasite Immunology), and is also pointed out by the reviewer in point 2 below.

2) Second, the authors use fixed values for the diagnostic performance (sensitivity and specificity) of the tests (skin snip microscopy for onchocerciasis and blood sample microscopy for LF), regardless of how the sensitivity may vary with the intensity of infection. The assumed values do not seem motivated by literature; at least a reference is not given, and I believe that the specificity could be higher (the microfilariae of the different skin-dwelling or blood-dwelling filariae are identifiable), but the sensitivity will vary with microfilarial load. At least in onchocerciasis, there will be an aggregated distribution of microfilariae in the skin and it has been shown by Bottomley et al. (2016, Parasit Vectors) that sensitivity varies with infection intensity, time after the last treatment, assumed impact of treatment on female worm microfilarial production and number of snips taken. I doubt that the sensitivity of the diagnostic will remain 95% in near elimination settings.

Because the sensitivity and specificity of the diagnostic tests (blood smear for LF and skin snip microscopy for onchocerciasis) are not well established, we chose arbitrary values of 0.95 to demonstrate the use of the freedom from infection probability calculations. We did not find consensus on these values in the literature (for LF: Chandrasena 2002, Irvine 2016, Weil 1997; for oncho: Boatin 2002, Bottomley 2016, Taylor 1989, Vlamnick 2015). For this reason, we indicated in both Methods (lines 442-445) and in the Results that at this stage the results are not definitive (line 228). We also conducted a sensitivity analysis to show that the identification of these values is important (Fig 2) and highlighted this limitation in our study and need for clarification on this issue in the discussion. We have added a sentence of clarification and some references for sensitivity/specificity values in the manuscript where we first mention the use of 0.95 for both metrics (line 226). The reviewer is correct in saying that the sensitivity of the tests may vary with the intensity of infection. At this time, there is little data/guidelines as to how to represent this in these calculations and so this will be a consideration for future work.

In Figure 2, it is not very clear how the diagnostic performance characteristics relate to sample size, but is likely that specificity will be very high, whilst sensitivity will likely be a lot lower than the lower value of 80% displayed.

We apologize that the relationship between diagnostic performance, sample size, and power of a survey to declare infection freedom was not explained clearly in this figure. We have added text to the figure caption to clarify what is being shown. The focus of this figure was to show that the freedom from infection results are highly dependent on diagnostic performance, and therefore this is a crucial area of research given that there is no clear consensus on these values. We also wanted to demonstrate the importance of sample size. In the dotted line, we show that, even if specificity is high, the power of the survey can be maintained by increasing the sample size (as opposed to the decrease in power shown by the solid lines where the sample size was kept constant). The sample sizes for this result are given in the figure caption.

3) Last, but not least, the generic filariasis model that the authors use is a lymphatic filariasis model

(originally developed by Norman and co-workers) presented elsewhere, and it is not applicable to onchocerciasis. The reasons for this are several:

We have edited the manuscript to explain that by “generic” we mean a vector-borne macro-parasitic disease that can be described by a structurally similar immigration-death model for describing linked infection dynamics in both the human and vector host (see line 448-451). Conceptualizing this as a generic model gives us the flexibility to specifically describe the infection of LF and onchocerciasis without presenting two separate complex models. The generic model is extended differently to describe LF or onchocerciasis transmission dynamics to make it specific to each parasite which is highlighted in Table S1. As noted in the text, the models are structurally similar, but we cite the key differences, namely 1) the inclusion of larval death rate in the onchocerciasis model, 2) the inclusion of excess vector mortality due to infection in the onchocerciasis model (this was missed in the previously submitted manuscript, so we thank the reviewer for prompting us to revisit this section!), and 3) the representation of the number of bites per vector as the human blood index divided by the gonotrophic cycle in the onchocerciasis model. We believe that our model is as applicable to onchocerciasis as previously developed models (as denoted by fits to data in Figure 1 and results in the Supplementary Material). Note given the complexity of the system being modelled, models of different structures may fit the normally sparser data equally well, a phenomenon called equifinality (Beven 2009, Poole and Raftery 2000). Thus, we cannot distinguish and say that one model is significantly better than another as that would require an infinite amount of data (see Beven 2009 and a wide literature on this problem), the reason why increasingly multi-model ensembles are used to combine all model predictions for a particularly application (see Smith et al. 2017).

3.i) The assumption of a constant mortality rate of adult worms (parameter μ sub W) implies an exponential distribution of adult worm survival times that is at odds with observations on microfilarial prevalence and intensity made after interruption of transmission under large-scale vector control (Plaisier et al. 1991 Acta Trop). These observations have led to modifications of onchocerciasis models by other authors (Walker et al. 2017 Epidemics). In onchocerciasis, treatment with ivermectin is not strongly and fast acting macrofilaricidal, so assumptions on the survival times of the adult worms do matter (they matter a lot less for LF, as the combination of ivermectin and albendazole seems to have a more pronounced effect on the survival of adult *Wuchereria bancrofti*).

We thank the reviewer for highlighting this point. We are aware of the ongoing discussion about which distribution best describes the survival of adult *Onchocerca volvulus* worms. At this stage, we have followed examples from Basáñez and Boussinesq 1999 (Phil. Trans. R. Soc. Lond.) and Filipe et al. 2005 (PNAS). We will address this issue more fully in future work and have added a statement regarding this limitation in the Methods section describing the models (line 471).

3.ii) The assumptions that immunity depends on mean adult worm load (equation for state variable I), and the operation of acquired immunity dependent on adult worm load (that enters into the equation for pre-patent infection, P) are not substantiated. In the Supplementary material, the authors tabulate parameters for immunosuppression affecting parasite establishment within humans (a hypothesis presented by Duerr et al. Trans R Soc Trop Med Hyg). However, other authors have argued that the different shapes of microfilarial intensity (not prevalence, which is a lot less informative given the nonlinear relationship between microfilarial prevalence and intensity) versus host age are due to differences in exposure (with age and among the sexes), without the need of invoking immunosuppression (Filipe et al. 2005, PNAS).

With regard to acquired immunity, our model indicates that immunity acts against incoming L3 as proposed by Duerr et al. 2003. The parameter for strength of acquired immunity is lower than those for LF to reflect the fact that this is likely not playing as significant a role in onchocerciasis.

With regard to immunosuppression, we consider the potential effects of both age-specific exposure and immunosuppression on parasite establishment in humans, and so consider the hypotheses proposed by both Duerr et al. 2003 (immunosuppression) as well as Filipe et al. 2005 (age and sex dependent exposure). Note again we do not make a priori decisions about the roles of either/both these types of immunity (lines 474-479); rather using our Bayesian approach, we essentially allow the data to inform the model, leaving open the possibility for one or both processes to occur.

3.iii) As I understand the parameters used from the Supplementary table, the authors have plucked a number of parameter ranges from the modelling literature, i.e. from a number of modelling studies that have tested different hypotheses about parasite population regulation within the human host, where we have most uncertainty. By doing so, they are implicitly assuming that the parameters are uncorrelated among themselves and independent of structural assumptions. This is not the case. In fact, and for instance, the proportion of infective larvae that establish within the humans is taken from models presented by Basáñez et al. (2002, *Am J Trop Med Hyg*) and Filipe et al. (2005, *PNAS*), but these parameters were estimated from fitting the models to data purposely ignoring immunosuppression. In turn, the parameters of immunosuppression were estimated by Duerr et al. ignoring the role of exposure. Hence, if the model the authors are using has both sets of mechanisms, exposure and immunosuppression, surely the parameters would have to be re-estimated from data and the hypotheses of either mechanism or both tested as structural and parametric assumptions given the importance of the exposure function. I don't think that using an 'off-the-shelf' LF model and using parameter sets that have been estimated with very different onchocerciasis models is the correct approach.

By using a Bayesian Melding approach (see Methods), we fully acknowledge the uncertainty in the model parameters and do not fix any parameter value in advance. Instead, we first assign prior parameter ranges based on values that have been published in the literature. Although the published values come with their own assumptions as the reviewer points out, this is the best knowledge we have of the system and they provide valuable information which allows us to constrain the parameters to reasonable ranges. Next, we sample 200,000 parameter vectors and run the model to calculate the corresponding 200,000 outputs (in this case, mf prevalence). The model outputs are then judged against the site-specific infection data and assigned a likelihood weight. The parameter vectors are finally resampled to select the most likely parameter vectors for the given site. In this way, all parameters are estimated from the site data. The sources given in the Supplementary Table S1 represent the available information we considered when assigning the prior parameter ranges. The parameter vectors vary from site to site and we rely on the data to choose the best-fitting vectors.

3.iv) The F functional forms for density-dependent processes are given in the Supplementary tables. In F3, for instance, one can envisage a combination of mating probability (positive density dependence) and density-dependent worm fecundity (negative density dependence), but this is not very clear and has different dynamic implications. In fact, another paper by Duerr et al. (*Int J Parasitol*) invokes density-dependent adult worm fecundity in onchocerciasis savannah settings. The existence of a breakpoint parasite density is dependent on the operation of positive density dependence. The authors state that function F4 contains larval mortality in onchocerciasis, but do not explain how the functional form may differ between *Simulium damnosum* and *S. neavei*. In the former, for instance, the functional form of

microfilarial uptake (limitation) and fly survival (decreasing with microfilarial intake) are important as described in other papers (reviewed by Basáñez et al. 2009, Adv Parasitol), but it is not clear what exactly goes into F4 for each simuliid species and how this is parameterised, even when referring to the Supplementary tables.

We are not very clear what issue is being raised in the first part of this comment. In equation F3, we describe the mf production in the human host which includes a positive density dependence in worm mating probability (the subfunction $\phi[W(a,t),k]$ which is described as a separate entry in Table S2). Worm fecundity is captured by the constants $s*\alpha$, where s is the proportion of worms reproducing and α is the production rate of mf per worm. In equation F4, we describe the larval density in the vector which includes a larval death rate, excess mortality due in the vector due to infection, and facilitation/limitation mf uptake functions which vary by species. We are sorry that the parameterization of the model is not clear. The parameters for larval death and excess mortality are given the same prior range regardless of species and they are fitted to infection data along with the other parameters as described in the Bayesian Melding explanation above in point 3.iii. For clarification about which function forms for mf uptake are used for each species, we have added some text to Supplementary Table S2.

3.v) In Africa, and for LF, vectors such as Anopheles species do have a cibarial armature (as opposed to Culex), but there are no onchocerciasis vectors with cibarial armature in Africa. This is not clearly stated, so it is difficult to understand which functional form goes with what exactly. This makes it difficult for other researchers to reproduce the work.

We have clarified in Supplementary Table S2 which function form is used for which mosquito and fly species.

In conclusion, I think the authors present an interesting approach but should focus solely on lymphatic filariasis, for which they have developed and parameterised a model that is specifically tailored to this disease.

We believe it is important to show both diseases to highlight the general applicability of this approach to more than just LF. This could be useful for other helminth NTDs, as well, such as STHs and schistosomiasis. We appreciate the issues raised by the reviewer with regard to our onchocerciasis model, but we are confident that our data-driven Bayesian approach considers the uncertainties and competing hypotheses discussed here. As we have pointed out in the paper, the focus here is not on the models but rather on the novel framework for coupling model predictions with surveillance data for making programmatic decisions. Currently, model predictions and surveillance data are considered separately, and we hope that this work will encourage more work to integrate the two and encourage collaboration between field teams and computational researchers involved in NTD control and elimination.

Minor comments:

4) I found reading the Abstract that it was not very informative of what precisely the paper is about. Using clearer, more concise language would be helpful to the reader.

We have modified the abstract to more clearly highlight the purpose of the paper.

Some sentences are very long and difficult to read, or lack punctuation, and would benefit from rephrasing and dividing into smaller, more digestible sentences. For instance: “Coupling infection survey-based metrics with parasite transmission dynamics model predictions of extinction thresholds can maximise the utility of information related to the potential infection status of a population to support objective calculations of the probability of achieving infection freedom.”

And at the beginning of the Results section: “Figure 1 provides an illustration of the ability of our modelling approach for not only learning best-fitting filarial models for observed or derived age-stratified Mf prevalence data in different community settings, but also for estimating Mf transmission thresholds for use as design prevalences based on the discovered best-fitting models or parameter vectors from each site.”

“Results are shown for two of the Onchocerciasis and one of the LF study sites in the figure, with the full set of model fits and the corresponding 95% Mf elimination thresholds (95% EPTs) at either the Annual Biting Rate (ABR) or the Threshold Biting Rate (TBR) values derived using the inverse cumulative density function (CDF) approach in all the other present study sites (depending on whether a particular location has implemented vector control measures in addition to Mass Drug Administration (MDA)) given in Supplementary Figures S1 and S2.”

These are only examples, but I found the language to be very verbose; sentences should be shorter, more concise and made clearer to the reader.

We have edited the manuscript to shorten sentences and use more concise language.

5) The definitions and goals of eradication and elimination are different and should not be used interchangeably.

By the same token, the definitions of infection and disease are different and should not be used interchangeably. Elimination of disease (presumably morbidity) is very different from elimination of infection.

We have edited the manuscript to ensure the appropriate terms have been used. This methodology can be applied in disease elimination or infection elimination contexts, which is why both terms are used throughout the manuscript. We have paid specific attention in this revision that we refer to infection when discussing our case examples of LF and onchocerciasis.

6) Some references have been left as EndNotes (e.g. {6 Sunish, 2002 #50}).

We have edited the manuscript to ensure the EndNote references are properly formatted.

7) Spelling of program and programme is inconsistent; it should be the latter.

We have edited the manuscript to use consistent spelling of programme.

8) Maximum acceptable disease prevalence or design prevalence should be defined from the outset.

We have added clarification in the introduction where these terms are first used (line 79), this is simply the breakpoint threshold prevalence (line 77).

9) Spell out all abbreviations/acronyms when first used (e.g. WHO, EPT).

We have made the necessary edits to ensure abbreviations are fully spelled out when first used.

10) There is a number of typos. Data is a plural noun.

We have revised the manuscript for typos and grammatical errors.

11) References need formatting according to journal style.

We have checked the reference formatting to ensure each one follows the appropriate style.

REVIEWERS' COMMENTS:

Reviewer #1 (Remarks to the Author):

This reviewer is satisfied with the authors' responses to his comments

Reviewer #2 (Remarks to the Author):

The authors are to be congratulated for having improved the clarity of the manuscript a great deal. I still would like to see a section on the limitations of the transmission models used, which are well discussed in the reply to referees letter but not in the paper. Therefore, I recommend that such a section be added in the Discussion of the paper.

I also attach an annotated version of the manuscript.

Subject: Substantiating freedom from parasitic infection by combining transmission model predictions with disease surveys (NCOMMS-18-08763B)

REVIEWERS' COMMENTS:

Reviewer #1 (Remarks to the Author):

This reviewer is satisfied with the authors' responses to his comments

We thank the reviewer for the valuable feedback provided and appreciate the recognition of the value of our work.

Reviewer #2 (Remarks to the Author):

The authors are to be congratulated for having improved the clarity of the manuscript a great deal. I still would like to see a section on the limitations of the transmission models used, which are well discussed in the reply to referees letter but not in the paper. Therefore, I recommend that such a section be added in the Discussion of the paper.

I also attach an annotated version of the manuscript.

We thank the reviewer for the feedback on our work, especially on the modeling methodology, and appreciate the opportunity to clarify and refine our manuscript. We have added content from the referee letter regarding limitations of the transmission model to the Discussion section of the manuscript as suggested. We respond to the comments left in the annotated manuscript below.

Comment 1: What does sustainable mean? If prevalence is sustainable it will not go to elimination?

Here, as described in the relevant paragraph (p4-5 in the tracked ms) we mean that, in contrast with an arbitrarily defined threshold which may result in recrudescence after interventions are stopped, the design prevalence needs to be defined at a value (ie the breakpoint value= design prevalence) that signifies the eventual attainment of zero prevalence rather than a prevalence will be sustained at this level or lower (because the arbitrarily set threshold is not the actual breakpoint threshold). This is explained fully if one reads the whole of paragraph 3 in the ms.

Comment 2: Perhaps then lower sensitivities should be tested in a sensitivity analysis given that the sensitivity of skin snips will be lower near elimination settings (Bottomley et al. 2016).

We have discussed limitations with regard to the diagnostic sensitivity and specificity in both the Results and Discussion sections and cited Bottomley et al.

Comment 3: But transmission breakpoints rely on the existence of positive density dependence processes that give rise to unstable equilibria. Do these processes operate in all systems? And how are

they balanced by negative density-dependent processes that may also operate and low infection values? This warrants discussion.

Here, we are making a general statement about the novelty of this freedom from infection framework which differs from the traditional use in that we consider non-zero thresholds. Non-zero thresholds are applicable to all macroparasitic infections and to many microparasitic infections with backward bifurcations.

Comment 4: This warrants a comment as to the parity status of the samples; if for some reason more nulliparous than parous mosquitoes were sampled, say, because the sampling method preferentially collects mosquitoes of different reproductive status, the sample size would need to be adjusted.

While we agree that the details of sampling and sample size calculations would need to be considered before applying the proposed sequential sampling approach, we feel this level of detail is beyond the scope of this discussion.

Comment 5: This really should be the WHO 2012 Roadmap on NTDs. The London Declaration endorsed the roadmap, but it is the WHO which set the targets formally.

We have edited this statement to reflect that the WHO set the targets.